



# Unlocking the Potential of Melting Calorimetry: A Field Protocol for Liquid Water Content Measurement in Snow

Riccardo Barella[1,*], Mathias Bavay[2], Francesca Carletti[2], Nicola Ciapponi[1], Valentina Premier[1], and Carlo Marin[1,*]

[1]Institute for Earth Observation, Eurac Research, Viale Druso, 1 - 39100 Bolzano, Italy
[2]WSL Institute for Snow and Avalanche Research SLF, Davos, 7260, Switzerland
[*]These authors contributed equally to this work.

**Correspondence:** carlo.marin@eurac.edu

**Abstract.**

Melting calorimetry, a classic experiment often conducted in high school chemistry laboratories, holds significant untapped potential for scientific applications beyond its educational context. Traditionally, this technique has been applied to measure the liquid water content in snow using two different formulations: melting calorimetry and freezing calorimetry. In contrast to freezing calorimetry, which is considered as the reference method for measuring liquid water content, melting calorimetry has been perceived as prone to generate significant inaccuracies. This research revisits the formulations for both melting and freezing calorimeters to assess volumetric liquid water content in snow. By incorporating the calorimetric constant, we account for heat exchange with the calorimeter, a critical factor often neglected in melting calorimetry experiments. This paper identifies the most effective and least uncertain method for determining this constant. A central contribution of this work is the introduction of a framework for estimating uncertainty in volumetric liquid water content measurements, adhering to established guidelines for uncertainty expression. This novel framework allows us to revisit past mathematical analyses and demonstrate that melting calorimetry delivers reliable measurements with an uncertainty $0.25\%$ greater than freezing calorimetry. Notably, despite this slightly higher uncertainty, melting calorimetry offers significant practical advantages for field applications. Moreover, we show how the proposed uncertainty framework can be expanded beyond instrumental uncertainty and take into account also the variability from environmental factors and operators, providing a more comprehensive characterization of the uncertainty. By exploiting the proposed uncertainty framework, we finally conduct an in-depth analysis for the optimal tuning of the experimental parameters. This analysis culminates in a robust field protocol for melting calorimetry that transcends common-sense procedural guidelines. Strict adherence to this protocol will maximize measurement accuracy. Applied in field tests in Italy and Switzerland, the melting calorimetry demonstrated to accurately tracking the wet front penetration in the snowpacks, producing results comparable to independent dielectric measurements. These findings highlight the accuracy and the practical advantages of melting calorimetry as a reliable field tool for quantifying snowpack liquid water content. Melting calorimetry can potentially serve as a valuable tool for independent calibration and validation of proximal and remote sensing techniques used for liquid water content retrieval.



**Table 1.** List of Symbols

| SYMBOL | Definition |
|---|---|
| $\theta_w$ | Snow Liquid Water Content expressed as percentage of liquid water for snow volume |
| $\theta_w^M$ | $\theta_w$ obtained with melting calorimeter |
| $\theta_w^F$ | $\theta_w$ obtained with freezing calorimeter |
| $V_s$ | Volume of the snow sample |
| $M_s$ | Mass of the snow sample |
| $M_i$ | Mass of the ice in the snow sample |
| $M_{W_{\theta_w}}$ | Mass of the liquid water fraction of the snow sample, it can be expressed as: $M_{W_{\theta_w}} = \theta_w M_s$ |
| $M_w, M_o$ | Mass of the melting and of the freezing agent, respectively |
| $M_{\mathrm{cal}}$ | Calorimeter internal vessel mass |
| $R$ | Ratio between $M_w$ and $M_s$ |
| $E$ | Calorimetric constant expressed in equivalent water mass |
| $T_w, T_o$ | Hot water and freezing agent initial temperature |
| $T_f$ | Final temperature of the system at the end of the experiment |
| $T_{mi}$ | Temperature of the ice cube after it is melted, and it is equal to 273.15 K |
| $T_{cv}$ | Temperature of the cold water employed for indirect calorimetric constant estimation |
| $T_s$ | is the temperature of the snow sample, that by definition is set to 273.15 K |
| $C$ | Water specific heat i.e., $4.2 \times 10^3$ J kg$^{-1}$K$^{-1}$ |
| $C_i$ | Ice heat capacity i.e., $2.09 \times 10^3$ J kg$^{-1}$K$^{-1}$ |
| $C_o$ | Freezing agent specific heat. In the case of using silicone oil, $C_o$ is $1.83 \times 10^3$ J kg$^{-1}$K$^{-1}$ at $-10°C$. |
| $C_{\mathrm{cal}}$ | Calorimeter internal wall specific heat |
| $L$ | Latent heat of ice fusion i.e., $3.34 \times 10^5$ J kg$^{-1}$ |
| $\rho_w$ | Water density i.e., $1000$ kg m$^{-3}$ |
| $\sigma_{\theta_w}$ | Uncertainty associated to the liquid water content measurement |
| $\sigma_{M_w}, \sigma_{M_w}$ | Uncertainty associated to the mass of melting and freezing agent, respectively |
| $\sigma_{M_s}$ | Uncertainty associated to the mass of the snow sample |
| $\sigma_{T_w}, \sigma_{T_o}$ | Uncertainty associated to the measure of the melting and freezing agent, respectively |
| $\sigma_{T_f}$ | Uncertainty associated to the final temperature of the system |
| $\sigma_{V_s}$ | Uncertainty associated to the volume of the snow sample |
| $\sigma_E$ | Uncertainty associated to the calorimetric constant |



# 1 Introduction

The presence of liquid water has a profound impact on the physical characteristics of snow, including heat advection through preferential flow, thermal conductivity, density, and mechanical properties, consequently influencing its hydrological and stability responses (Techel and Pielmeier, 2011; Avanzi et al., 2017; Wever et al., 2016; Moure et al., 2023). Therefore, the precise measurement of liquid water content ($\theta_w$) within the snowpack assumes critical importance as it provides invaluable information to properly describe the current conditions of the snowpack and predict its evolution (Hirashima et al., 2019;

Wever et al., 2014). Moreover, a proper characterization of liquid water content in the snowpack is essential to characterize the backscattering of radar signal e.g., (Gagliano et al., 2023; Marin et al., 2020).

Calorimetry is the scientific technique used to measure the heat energy transferred during a physical or chemical process, such as a reaction or a state change. This is achieved by utilizing a calorimeter, a specialized device designed to accurately measure the heat exchanged between a system and its surroundings. It has emerged as a promising technique for $\theta_w$ deter-

mination within the snowpack, e.g., (Yosida, 1960; Jones et al., 1983; Kawashima et al., 1998; Jones, 1979; Boyne and Fisk, 1987; Kinar and Pomeroy, 2015). To measure the $\theta_w$ in snowpacks, calorimetry offers two distinct approaches based on the process involved: melting and freezing calorimetry. In a melting calorimeter, a snow sample is immersed in hot water. This results in the transition of the solid portion of the sample to a liquid phase, and the heating of the melted ice portion and $\theta_w$ to the equilibrium temperature. The measurement of the energy required for this transition is directly related to the amount of ice present in the snow sample. Consequently, $\theta_w$ can be derived as the difference between the mass or the volume of the sample

and the ice content. This is why melting calorimetry is considered an indirect measurement (Colbeck, 1978). On the contrary, a freezing calorimeter involves immersing a snow sample in a freezing agent such as cooled silicon oil that induces the transition of any liquid water in the sample to a solid phase, and the cooling of the ice fraction and of the freezed $\theta_w$ to the equilibrium temperature. The measurement of the energy required for this transition is directly related to the amount of $\theta_w$ present in the

snow sample. Freezing calorimeter is generally accepted as a reference standard for measurements (Colbeck, 1978) and it was used in the past to calibrate and validate non-destructive methods for liquid water content measurements (e.g., Denoth et al. (1984); Stein et al. (1997); Kendra et al. (1994)).

The selection of the most suitable approach for implementing calorimetry demands consideration of both field usage and the accuracy of the obtained results. Firstly, it becomes evident that the practical handling of these methods significantly varies.

Specifically, the usage of freezing calorimeter presents several challenges. This calorimeter requires the use of a freezing agent such as silicone oils or toluene, which possess characteristics that make them less desirable for use. For instance, these agents may be toxic and pose difficulties in terms of proper disposal and cleaning of the instruments after use. Due to the variability of the employed freezing agents, its specific heat has to be retrieved through a dedicated analysis every time a new agent is used. Furthermore, operating a freezing calorimeter poses challenges associated with maintaining the freezing agent within a

temperature range of $-50$ to $-20°C$. This task is complicated by thermal losses and operational difficulties encountered in cooling down the active agent on the field. Additionally, active monitoring of temperature changes is required throughout the



experiment, lasting at least 15 minutes (Jones et al., 1983), to ensure complete freezing of $\theta_w$ and the system reaching thermal equilibrium. During this period, thermal losses can be substantial, introducing significant uncertainties.

In melting calorimetry, keeping the melting agent warm in a cold environment remains an issue. However, this can be mitigated by using a portable stove to raise water temperature. In contrast to freezing calorimetry, a large mass of ice needs to be melted, requiring a considerable amount of energy. Despite this, the process, with proper mixing, can achieve thermal equilibrium quickly, resulting in fewer potential thermal losses through the calorimeter. This enhances the overall efficiency of the experiment compared to a freezing calorimeter.

Regardless of being the established benchmark for measuring $\theta_w$, the freezing calorimeter limitations have stimulated the development of alternative methods. These include, among the most established and used, the alcohol calorimeter, which utilizes methanol (Fisk, 1986) and the dilution technique, which employs a 0.01 N hydrochloric acid stock solution (Davis et al., 1985). Laboratory comparisons have demonstrated that these two methods yield equivalent results to the freezing calorimeter and the denothmeter (Boyne and Fisk, 1987), a portable instrument that measures liquid water content in snow using its dielectric properties (Denoth and Foglar, 1986). The advantage of the dilution technique lies mainly in its speed compared to the freezing calorimeter. Following its successful comparison with the freezing calorimeter, the dilution technique was used to gather extensive data for calibrating and validating various dielectric models (Perla and Banner, 1988; Perla, 1991). These studies revealed limitations in the models, also in the one used by the denothmeter. Indeed, despite performing well against other methods (Boyne and Fisk, 1987), this model is accurate only for low $\theta_w$ values (Denoth et al., 1984; Perla, 1991). The high variability in wet snowpacks is a key factor in this discrepancy. As liquid water content increases, water intrusions change shape, eventually saturating the air spaces. Moreover, ice grains cluster and melting - refreezing cycles can create large ice structures like ice pipes. These changes in the snowpack internal composition lead to a diverse interaction with the electromagnetic field, depending on the specific snowpack conditions (Colbeck, 1980; Camp, 1992). Consequently, identifying a universally accepted optimal dielectric, or in general permittivity model remains a challenge, necessitating further research (Picard et al., 2022). A crucial step towards a more accurate model is collecting a large amount of field measurements. In this sense, melting calorimetry offers several advantages, not only over freezing calorimetry but also over alcohol and dilution methods: i) it eliminates the need to prepare a specific stock solution; ii) it has less stringent temperature requirements for the solution (it does not need to be precisely at $0°C$); and iii) it allows for easier mixing. Interestingly, despite these advantages, melting calorimeters have not been widely adopted in the past.

The fact that the melting calorimeter was avoided is likely due to the perception of large errors associated with it (Colbeck, 1978; Fisk, 1986; Boyne and Fisk, 1987; Denoth and Foglar, 1986). In particular, the study conducted by Colbeck (1978), where several measuring techniques including the freezing and melting calorimeter were compared within a theoretical framework of uncertainty propagation, identifies the melting calorimeter as "inherently inaccurate". The primary objective of Colbeck (1978) was to determine the measuring methodology that would result in lower uncertainty when deriving the water saturation and porosity. The analysis revealed that the uncertainty is propagating in a larger quantity when starting from a measurement of ice volume, i.e., utilizing the melting calorimeter compared to starting from a measurement of water volume, i.e., employing the freezing calorimeter. Based on this analysis, assuming that the relative uncertainty in measuring the ice volume is similar or





greater than that in measuring the water volume, the freezing calorimeter was deemed preferable over the melting calorimeter. However, the study provided only an intuitive explanation to support this assumption without actually calculating the relative uncertainties of ice and water, suggesting that freezing a smaller amount of water leads to a reduced error compared to melting

a larger amount of ice. Nonetheless, this overlooks the practical challenges associated with freezing a small amount of water within a snow sample under real-world conditions, as discussed previously. Despite Kawashima et al. (1998) attempt to address Colbeck (1978) criticisms of melting calorimetry, their work suffers from limitations. Firstly, they do not account for heat exchange with the calorimeter. This issue has been recognized since the early applications of melting calorimeter (Halliday, 1950) and has been recently re-emphasized by Fasani et al. (2023). Secondly, their mathematical uncertainty propagation

probably relies on simplified assumptions. These shortcomings likely compromise the accuracy of $\theta_w$ measurements obtained using melting calorimeter. Furthermore, these limitations have already propagated into subsequent studies like (Fasani et al., 2023; Webb et al., 2022; Mavrovic et al., 2020), potentially invalidating their findings. Therefore, while the field applicability of the melting calorimetry is attractive, a critical reevaluation of this technique is necessary. This should focus on the mathematical formulation and uncertainty propagation to determine the true potential of the melting calorimetry for $\theta_w$ measurement and its

applicability as a reference measurement.

Our paper addresses and rectifies the limitations of the state of the art for calorimetric analysis for liquid water measurement by proposing a sound mathematical formulation. This provides a more rigorous understanding of the technique accuracy so that the application of melting calorimetry in the future is correct and sound from the critics. In this sense, this paper prioritizes unlocking the full potential of the melting calorimeter for $\theta_w$ measurement, rather than providing a comprehensive review of

all existing measurement methods. Moreover, while certain concepts may resemble those in previous works in broad terms, our formulation and its application differ significantly, constituting the primary novelty of our paper. In detail, the novelties lie in the following points

– We present the formulations for both melting and freezing calorimeters, it is expressed in %vol and incorporating the calorimetric constant.

– We clarify the most effective method for estimating the calorimetric constant ($E$).

– We introduce the formulations for estimating volumetric liquid water measurement uncertainty for melting and freezing calorimetry, emphasizing its importance for facilitating comparisons between measurements. We do not account not only for instrument uncertainty but we also include the random uncertainty introduced by different environment conditions and operators. Something that was never attempted in the past.

– While Colbeck (1978) argued that melting calorimetry is "inherently inaccurate", a view echoed in later studies, we revisit and refine this analysis. This mathematical refinement, occurring 45 years after Colbeck's original paper, fundamentally alters the prevailing understanding that the melting calorimeter, an indirect measurement of liquid water content, is not inherently inaccurate, a significant breakthrough that we believe researchers in the field will appreciate.

– we have devised a field protocol that effectively minimizes the uncertainties.



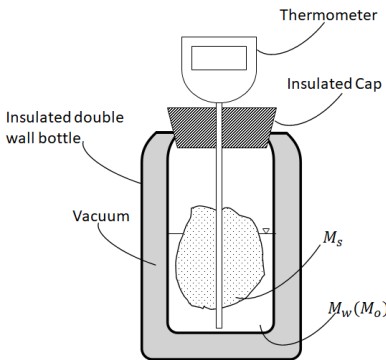

**Figure 1.** The figure illustrates a cross-section of the calorimeter, depicting a schematic representation of its components. In detail it is possible to notice the double wall insulated bottle and the vacuum in between the walls, a snow sample ($M_s$) immersed in the melting ($M_w$) or freezing agent ($M_o$). For convenience, the thermometer is embedded within the insulated cap of the system.

– We have made available all the codes necessary for calculating liquid water content and its associated uncertainty based on the measured variables.

Regarding the field protocol, it is termed as the "field protocol" but it transcends commonsense procedural guidelines for conducting analyses in real-world settings. Rather, it represents the culmination of comprehensive mathematical development presented within the paper. The protocol includes specific instructions on the amount of hot water to be used, its temperature,

the size of the calorimeter, the masses involved, and other crucial details for controlling the uncertainty during the experiment replication. While certain instructions may resemble those in (Kawashima et al., 1998), our formulation and the mathematical justification differ significantly. The proposed protocol was applied in two different test sites in Italy and Switzerland by two different research groups with different melting calorimeters. The results, compared to independent measurements of dielectric constant of the snow using the Denothmeter, show how the application of the proposed protocol to the melting calorimeteric

measurements is able to properly track the wet front penetration inside the snowpack in an accurate way.

## 2   Formulation of Melting and Freezing Calorimetry

The calorimeter experiment is formulated as an energy budget problem. Each term in the energy budget associated with a change in temperature depends on the mass involved, their specific heat, and the temperature difference. Conversely, terms related to phase transitions are determined by the mass involved and the latent heat. Before dealing with the computation of

the energy balance, some practical considerations on the setup and measurement operations will be provided.

The setup for calorimetry used to measure $\theta_w$ in snow involves three essential instruments: i) a specialized instrument known as a calorimeter (see Fig. 1), ii) a scale, and iii) a thermometer. The calorimeter is designed as an insulated container to maintain a given temperature and create an adiabatic environment, ensuring ideal heat exchange between the snow sample and the melting or freezing agent. The thermometer is generally already part of the calorimeter. It is inserted into the insulated





cap of the calorimeter, with particular attention to not introducing any points for heat exchanges, and it is used to monitor

temperature changes during the experiment. Prior to the wide availability of modern materials and construction techniques,

Yosida conducted extensive research on optimizing calorimeter design (Yosida, 1960). Today, calorimeters can be as simple

as a commercially available insulated bottle, which provide superior thermal insulation than the Yosida design. The vacuum

between the vessels significantly reduces heat transfer by both conduction and convection. Additionally, silvered internal walls

minimize heat loss through thermal radiation. These advancements extend the time during which the calorimeter can work as

an adiabatic environment, eliminating the need for the double-container design (one for hot water and one for snow) and the

stirrer introduced by Yosida (1967).

From a practical point of view, the calorimetric experiment starts with the placement of a precise mass of either melting agent

($M_w$) or freezing agent ($M_o$) into the insulated container. The most commonly used melting agent is hot water, while the most

used freezing agent is silicone oil. The initial temperature of the melting agent ($T_w \gg 0°C$) or the freezing agent ($T_o \ll 0°C$) is

recorded. Subsequently, a snow sample of mass $M_s$ and volume $V_s$ is added to the calorimeter. The resulting mixture is stirred

to facilitate rapid mixing and efficient heat transfer. During this process, the temperature of the mixture is monitored until it

reaches equilibrium ($T_f$), indicating the completion of the heat exchange between the snow sample and the agent. Minimizing

the duration of these steps is essential to maintain the assumption of an adiabatic system. Any potential losses of the calorimeter,

which may be large in the freezing calorimeter giving the longer time of operation, must be accounted for in the analysis e.g.,

monitoring the temperature for long time (Jones et al., 1983). Calorimetric experiments involve heat exchange between the

snow sample and the fluid. During the calorimetric experiment, heat exchange occurs not only between the snow sample

and the fluid, but also with the internal metallic calorimeter wall. The wall, due to its high thermal conductivity and vacuum

insulation, rapidly reaches thermal equilibrium. While thermal radiative heat transfer represents the most significant energy

loss for the calorimeter, the impact of radiation can be considered negligible when compared to the timescale of the experiment

(around one minute). Therefore in this study, we will focus primarily on heat conduction as the dominant mechanism for heat

exchange as done also in Jones et al. (1983). All the mentioned quantities are then used to calculate the heat exchanged during

the process, allowing for an estimation of $\theta_w$ within the snowpack. As we progress with the paper, when we discuss liquid

water content, we will be specifically referring to the volumetric liquid water content (Fierz et al., 2009), which is a measure

defined as the ratio of the volume of liquid water content to the volume of the snow sample. Regarding the notation, $\theta_w^M$ refers

to the volumetric liquid water content measured with the melting calorimeter, while $\theta_w^F$ identifies the same quantity measured

with the freezing calorimeter as reported in Table 1.

From the heat exchange point of view, in the melting calorimeter, the energy introduced by the hot water $Q_{hot\ water}$ and the

calorimeter internal wall $Q_{calorimeter}$ must balance with the sum of energy terms for the sinks. These include the heat needed

for melting the ice content $Q_{ice\ melt}$ and the heat required to bring the melted ice and the liquid water already present in the

snow to the equilibrium temperature, $Q_{melted\ ice}$ and $Q_{\theta_w}$. In Eq.(1) is shown the energy budget, detailing the different terms

of the equations





$$\underbrace{Q_{hot\ water}}_{M_w C(T_f - T_w)} + \underbrace{Q_{calorimeter}}_{M_{\text{cal}} C_{\text{cal}}(T_f - T_w)} + \underbrace{Q_{ice\ melting}}_{LM_i} + \underbrace{Q_{melted\ ice}}_{M_i C(T_f - T_s)} + \underbrace{Q_{\theta_w}}_{M_{W_{\theta_w}} C(T_f - T_s)} = 0 \qquad (1)$$

Where:

– $C$ represents the specific heat of water ($4.2 \times 10^3$ J kg$^{-1}$K$^{-1}$);

– $L$ represents the latent heat of fusion of ice ($3.34 \times 10^5$ J kg$^{-1}$);

– $T_s$ is the temperature of the snow sample, that by definition is set to 273.15 K;

– $M_{\text{cal}}$ represents the mass of the internal wall of the calorimeter in contact with the water;

– $C_{\text{cal}}$ represents the specific heat of the internal wall of the calorimeter in contact with the water;

– $M_{W_{\theta_w}}$ is the mass of the liquid water fraction of the snow sample and can be expressed as:
$M_{W_{\theta_w}} = \theta_w M_s$;

– $M_i$ is the mass of the ice fraction of the snow sample and can be expressed as:
$M_i = M_s - M_{W_{\theta_w}}$;

In the operative formulation of the melting calorimeter, the heat exchange contribution of the calorimeter internal wall is
expressed as an additive term to the water mass, introducing the so-called calorimetric constant $E$ (Jones et al., 1983; Fasani et al., 2023). Therefore, as an inherent material property of the calorimeter, the computation of $E$ can be accomplished as follows

$$E = \frac{M_{\text{cal}} C_{\text{cal}}}{C} \qquad (2)$$

An accurate estimation of $E$ requires precise information about the calorimeter construction. Ideally, the manufacturer
can provide detailed specifications, including the weight of the internal container and the material used in its construction (typically stainless steel). As a last resort, a destructive approach can be employed. This involves carefully dismantling the calorimeter and precisely measuring the weight of the internal container. Several non-destructive methods for estimating $E$ have been proposed, such as mixing fluids at different temperatures (e.g., (Jones et al., 1983; Austin, 1990)) or inverting eq. (3) (e.g., (Fasani et al., 2023)), given that wet snow samples are prepared at a given $\theta_w$. However, these methods often
suffer from significant uncertainty propagation. For instance, applying the method used by Jones et al. (1983); Austin (1990) yields an uncertainty of $\sigma_E = 2.6$g for $E = 6.58$g. Similarly, creating artificial wet snow samples (Fasani et al., 2023) can further increase uncertainty to unacceptable levels i.e., $\sigma_E = 4.87$g, even if we assume that $\theta_w$ is known without uncertainty. Therefore, although non-destructive, these methods are associated with significant uncertainties and are thus recommended to





be avoided. The complete mathematical derivations for estimating the uncertainty associated with $E$ using various methods
from the literature are provided in Appendix A.

From Eq.(1) and Eq.(2) $\theta_w^M$ can be derived as follows:

$$\theta_w^M = \frac{M_s}{V_s \rho_w}\left(1 - \frac{C}{L}\left[\frac{(M_w + E)(T_w - T_f)}{M_s} - (T_f - T_s)\right]\right) \tag{3}$$

This formulation differs from Kawashima et al. (1998) in the inclusion of the parameter $E$, its simplification of the involved
masses, and its use of density for volumetric conversion (i.e., snow density $M_s/V_s$, and water density $\rho_w$).
The freezing calorimeter operates on a similar principle to the melting calorimeter, however in this case the freezing agent
and the container extract heat from the ice content and $\theta_w$ in the snow, causing it to freeze. The corresponding energy balance
equation is given by:

$$\underbrace{Q_{freezing\ agent}}_{M_oC(T_f - T_o)} + \underbrace{Q_{calorimeter}}_{M_{cal}C_{cal}(T_f - T_o)} + \underbrace{Q_{cooling\ ice}}_{M_iC_i(T_f - T_s)} + \underbrace{Q_{freezed\ \theta_w}}_{-M_{W_{\theta_w}}L} + \underbrace{Q_{cooling\ freezed\ \theta_w}}_{M_{W_{\theta_w}}C_i(T_f - T_s)} = 0 \tag{4}$$

Where, $C_o$ is the heat capacity of the freezing agent, and $C_i$ is the heat capacity of ice ($2.09 \times 10^3$ J kg$^{-1}$K$^{-1}$). In the case
of using silicone oil, $C_o$ is $1.83 \times 10^3$ J kg$^{-1}$K$^{-1}$ at $-10°C$.

- $C_o$ is the heat capacity of the freezing agent. In the case of using silicone oil, $C_o$ is $1.83 \times 10^3$ J kg$^{-1}$K$^{-1}$ at $-10°C$;

- $C_i$ is the heat capacity of ice ($2.09 \times 10^3$ J kg$^{-1}$K$^{-1}$);

As for the melting calorimeter, the heat exchange contribution of the calorimeter internal wall is expressed as an additive
term, but this time on the freezing agent mass $M_o$. The calorimetric constant $E^F$ in that case can be obtained as:

$$E^F = \frac{M_{cal}C_{cal}}{C_o} \tag{5}$$

In the freezing case, the $\theta_w^F$ is directly related to the temperature difference induced by the freezing of water present in the
snow and, as shown in (Jones et al., 1983), it is derived from Eq.(4) and Eq.(5) as follows:

$$\theta_w^F = \frac{M_s}{V_s \rho_w}\left(\frac{(M_o + E^F)C_o(T_f - T_o)}{LM_s} - \frac{C_i(T_s - T_f)}{L}\right) \tag{6}$$

## 3 Propagation of the Uncertainty in Calorimetry

In scientific measurements, accounting for uncertainty propagation is crucial to accurately quantify the uncertainty associated
with the obtained results (IEC et al., 1993; Moffat, 1988). Moreover, to assess measurement effectiveness and choose the



formulation yielding the lowest uncertainty, the analysis of uncertainty propagation should be the first step. As exemplified in the previous section for the calorimetric constant, various formulas exist for calculating $E$, but they differ significantly in handling instrumental uncertainty propagation. Analyzing uncertainty propagation beforehand would have readily identified
the optimal approach. Unfortunately, this crucial step is often overlooked.

The overall measurement uncertainty is influenced by a variety of factors, including instrumental uncertainties, environmental conditions and variations introduced by the operator during the experiment. Properly accounting and quantifying these sources of uncertainty is essential to ensure the reliability and validity of the $\theta_w$ measurements using calorimetry. While it is factored into the theoretical development, and generally it is illustrated with common values, it is crucial to apply the uncer-
tainty calculation for each new measurement. This additional information allows us to better interpret the results and compare different measurements. Previous attempts to quantify uncertainty in freezing and melting calorimeters relied only on instrumental uncertainty propagation. Other sources of error, such as operator variations and environmental factors, were often added to the instrumental uncertainty with subjective reasoning (and quantities), leading to inconsistent interpretations. As a result, a wide range of values, generally expressed by percentage of mass, from "several percent" (Colbeck, 1978; Linlor, 1975) to
$\pm 2\%$ for melting calorimeters (Kawashima et al., 1998) and $0.5\%$ to $1 - 2\%$ for freezing (Jones, 1979; Jones et al., 1983; Fisk, 1986), have been reported as absolute uncertainties. This paper aims to address this historical ambiguity and establish a more rigorous approach to uncertainty quantification in calorimetry for liquid water content estimation.

In Section 3.1, we will initiate the uncertainty propagation analysis by focusing on the instrument uncertainty. The losses occurring during the experiment realization can be assumed to be equal for both melting and freezing calorimeters since the
basic operations are analogous. This will enable a direct comparison between the melting and freezing calorimeters, revealing the suitability of the methods for real $\theta_w$ measurements. In Section 3.2, practical considerations about each step of the melting calorimetric experiment will be explored, focusing on minimizing all sources of uncertainty. Section 3.3 shows the results of the sensitivity analysis of the melting calorimeter providing all the indications for a field protocol that maximizes the accuracy. Finally, in Section 3.4, we will extend the analysis to include uncertainties due to operator and environmental variations in the
melting calorimetry, something never attempted before. This is done by conducting repeated measurements done by different operators and under different environmental conditions, which will also showcase both the method consistency and its accuracy.

### 3.1   Instrumental uncertainty propagation: Melting vs Freezing Calorimetry

Instrumental uncertainty, which arises from the limitations and imperfections of the measuring instruments used in the calorimetric experiment, propagates into the final estimation of $\theta_w$. In both melting and freezing calorimetry, the uncertainties are
associated with temperature measurements and mass determinations. To quantify uncertainty propagation, we employ a statistical method for uncertainty propagation in accordance with the Guide to the Expression of Uncertainty in Measurement (GUM) (IEC et al., 1993). In detail, the uncertainty $\sigma_{\theta_w^{M,F}}$ can be determined as the squared root of the sum of the squared partial derivatives of $\theta_w^{M,F}$ with respect to the variables with an associated uncertainty, each, multiplied by the squared associated error. By assuming independent variables, the general formulation is reported in Eq. (7) (IEC et al., 1993; Moffat, 1988).





$$\sigma_{\theta_w^{M,F}} = \sqrt{\sum_{m_i}\left(\frac{\partial \theta_w^{M,F}}{\partial m_i}\right)^2 \sigma_{m_i}^2} \tag{7}$$

In Eq. (7), $m_i$ represents a single measurement affected by uncertainty, and $\sigma_{m_i}$ is the associated uncertainty. For the melting calorimeter, the required measurements are $V_s, M_w, M_s, T_w, T_f$ and $E$ (see Eq. (3)) with associated instrumental uncertainties $\sigma_{M_w}$ and $\sigma_{M_s}$, which depends on the accuracy of the scale, $\sigma_{T_w}$ and $\sigma_{T_f}$, which depends on the accuracy of the thermometer, $\sigma_{V_s}$, which depends on the uncertainties in measuring the volume of the sampler; and finally on $\sigma_E$, which depends on the uncertainty of the estimation of $E$ (see Appendix A). In previous studies, different uncertainty estimators have been used like the sum of the relative uncertainty (Colbeck, 1978), or the sum of the absolute uncertainty (Jones, 1979; Kawashima et al., 1998) to calculate the uncertainty in the mass of liquid water. This paper breaks new ground by providing the volumetric uncertainty that adheres to the GUM guidelines.

Equation (7) can be applied to Eq. (3) and expanded as follows

$$\sigma_{\theta_w^M} = \sqrt{\left(\frac{\partial \theta_w^M}{\partial M_w}\right)^2 \sigma_{M_w}^2 + \left(\frac{\partial \theta_w^M}{\partial M_s}\right)^2 \sigma_{M_s}^2 + \left(\frac{\partial \theta_w^M}{\partial T_w}\right)^2 \sigma_{T_w}^2 + \left(\frac{\partial \theta_w^M}{\partial T_f}\right)^2 \sigma_{T_f}^2 + \left(\frac{\partial \theta_w^M}{\partial V_s}\right)^2 \sigma_{V_s}^2 + \left(\frac{\partial \theta_w^M}{\partial E}\right)^2 \sigma_E^2} \tag{8}$$

The partial derivatives in Eq. (8), known as sensitivity coefficients, are calculated as follows (to preserve the explicit dependence on snow density i.e., $\rho_s = M_s/V_s$, we refrain from simplifying terms involving $M_s$):

$$\frac{\partial \theta_w^M}{\partial M_w} = -\frac{M_s}{V_s \rho_w}\frac{C}{L}\frac{T_w - T_f}{M_s} \tag{9}$$

$$\frac{\partial \theta_w^M}{\partial T_w} = -\frac{M_s}{V_s \rho_w}\frac{C}{L}\frac{M_w + E}{M_s} \tag{10}$$

$$\frac{\partial \theta_w^M}{\partial T_f} = \frac{M_s}{V_s \rho_w}\frac{C}{L}\left(\frac{M_w + E}{M_s} + 1\right) \tag{11}$$

$$\frac{\partial \theta_w^M}{\partial M_s} = \frac{M_s}{V_s \rho_w}\frac{C}{L}\frac{(M_w + E)(T_w - T_f)}{M_s^2} + \frac{1}{V_s \rho_w}\left(1 - \frac{C}{L}\left(\frac{(M_w + E)(T_w - T_f)}{M_s} - (T_f - T_s)\right)\right) \tag{12}$$

$$\frac{\partial \theta_w^M}{\partial V_s} = -\frac{M_s}{V_s^2 \rho_w}\left(1 - \frac{C}{L}\left(\frac{(M_w + E)(T_w - T_f)}{M_s} - (T_f - T_s)\right)\right) \tag{13}$$

$$\frac{\partial \theta_w^M}{\partial E} = -\frac{M_s}{V_s \rho_w}\frac{C}{L}\frac{T_w - T_f}{M_s} \tag{14}$$




It is important to note that although there are significant differences in our calorimetric equations, Kawashima et al. (1998)
calculated some of the same derivatives we present. Here is a breakdown of the key distinctions.

The approximation used to transition from Eq. (1) to Eq. (4) in their original paper, via Eq. (3), is not clear. This ambiguity
results in different values when properly calculating the derivatives. Despite being mathematically sound, Kawashima et al.
(1998) propose fixing $T_f$ for uncertainty propagation.

These points highlight the differences and clarify the approaches taken in the respective works. However, as detailed in
Section 2, calorimetric experiments begin with hot water at a known $T_w$. The fixed-$T_f$ approach makes their uncertainty
propagation method non-representative of the propagation of the error under real conditions making Figure 3 of the original
paper misleading.

Similarly, for the Freezing Calorimeter, we can analyze the error propagation associated with temperature and weight mea-
surements of Eq. (6). In detail, by applying Eq. (7) we obtain

$$\sigma_{\theta_w^F} = \sqrt{\left(\frac{\partial \theta_w^F}{\partial M_o}\right)^2 \sigma_{M_o}^2 + \left(\frac{\partial \theta_w^F}{\partial M_s}\right)^2 \sigma_{M_s}^2 + \left(\frac{\partial \theta_w^F}{\partial T_o}\right)^2 \sigma_{T_o}^2 + \left(\frac{\partial \theta_w^F}{\partial T_f}\right)^2 \sigma_{T_f}^2 + \left(\frac{\partial \theta_w^M}{\partial V_s}\right)^2 \sigma_{V_s}^2 + \left(\frac{\partial \theta_w^F}{\partial E}\right)^2 \sigma_E^2} \quad (15)$$

The partial derivatives in Eq. (15) are calculated as follows:

$$\frac{\partial \theta_w^F}{\partial M_o} = \frac{M_s}{V_s \rho_w} \frac{C_o(T_f - T_o)}{L M_s} \quad (16)$$

$$\frac{\partial \theta_w^F}{\partial T_o} = -\frac{M_s}{V_s \rho_w} \frac{(M_o + E) C_o}{L M_s} \quad (17)$$

$$\frac{\partial \theta_w^F}{\partial T_f} = \frac{M_s}{V_s \rho_w L} \left(\frac{(M_o + E) C_o}{M_s} + C_i\right) \quad (18)$$

$$\frac{\partial \theta_w^F}{\partial M_s} = -\frac{1}{L} \left(\frac{(M_o + E) C_o(T_f - T_o)}{V_s \rho_w M_s} + \frac{1}{V_s \rho_w} \left(\frac{(M_o + E) C_o(T_f - T_o)}{M_s} - C_i(T_s - T_f)\right)\right) \quad (19)$$

$$\frac{\partial \theta_w^F}{\partial V_s} = -\frac{M_s}{V_s^2 \rho_w} \left(\frac{(M_o + E) C_o(T_f - T_o)}{L M_s} - \frac{C_i(T_s - T_f)}{L}\right) \quad (20)$$

$$\frac{\partial \theta_w^F}{\partial E} = \frac{M_s}{V_s \rho_w} \frac{C_o(T_f - T_o)}{L M_s} \quad (21)$$

Similar derivatives have been presented in (Jones, 1979) for the propagation of the instrumental uncertainty for the liquid
water content expressed in percentage of mass.





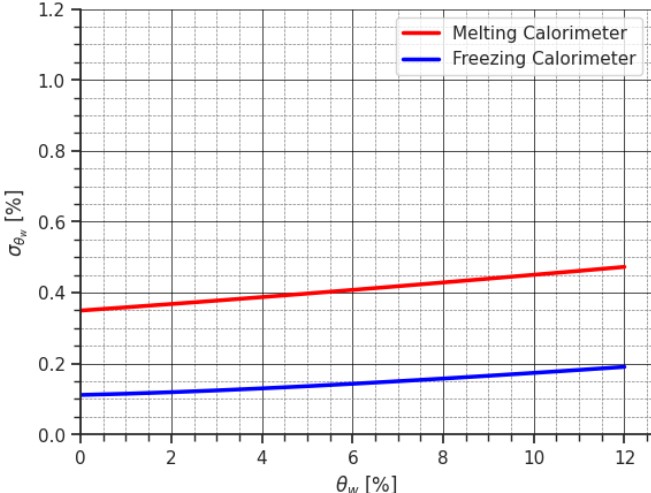

**Figure 2.** The figure provides a comparison of uncertainty in estimating $\theta_w$ by propagating instrumental uncertainty using both the melting and freezing calorimeter methods. Notably, freezing calorimetry demonstrates lower uncertainty. However, it is essential to recognize that both methods offer uncertainties that still enable accurate measurement of $\theta_w$.

By evaluating Eqs. (8) and (15) it is possible to compare the two methods as done in Colbeck (1978). In detail, by assuming $V_s = 200 \; \mathrm{cm}^3$, a value consistent with the density measurement sampler (Proksch et al., 2016), which ensures compact dimensions for both the snow sampler and the calorimeter, facilitating transportation, and considering a snow density of the dry snow of $366 \; \mathrm{kg\,m}^{-3}$ as a legacy of (Colbeck, 1978), to which we add a given percentage of liquid water increasing the density of consequence. We set the temperature and mass of hot water at $40°\mathrm{C}$ and 2 times $M_s$, respectively, in line with the analysis of

Section 4. Similarly, we set the temperature and mass of the freezing agent at $-30°\mathrm{C}$ and 1.3 times $M_s$, respectively, as per (Jones et al., 1983). The calorimetric constant was assumed to be equal to $6.58\mathrm{g}$. $\theta_w$ was varied from 0 to 12%, considering the most common values in melting snowpack. Additionally, we consider a scale with uncertainty of $0.1$ g, a thermometer with uncertainty $0.2°\mathrm{C}$, an uncertainty in the estimation of the calorimeter constant $E$ of $0.1$ g. Regarding the uncertainty associated with the sample volume, to the best of our knowledge, no studies have been conducted in that specific direction. However from

our practical experience, a value of $2 \; \mathrm{cm}^3$ can be realistic according to the tools we are using.

By substituting these values into Eqs. (8) and (2), we quantified the uncertainty associated to $\theta_w$. The results are illustrated in Fig. 2. The comparison demonstrates the superior performance of the freezing calorimeter, particularly for high $\theta_w$ values, where the melting calorimetry reaches almost $0.5\%$ uncertainty whereas the freezing calorimetry stops at $0.2\%$ uncertainty, a similar value obtained by Jones et al. (1983); Jones (1979) when converted to the sum of the absolute errors and in percentage of

mass (note that the sum of the absolute uncertainties is always larger than the root mean square of the uncertainties). However, the general low value of $\sigma_{\theta_w^M}$, indicates that the melting calorimeter can still provide a significant and reliable measurement of $\theta_w$. Coupled with its notable practical advantages, the melting calorimeter becomes an attractive and compelling choice for field applications, particularly in remote areas.



It is important to highlight that, the uncertainty analysis presented here produces the same results as the paper of Colbeck
(1978). By applying Eq. (8) for water saturation $S_w$, we can find that its relative uncertainty $\Sigma_{S_w}$ is 5.1 times the relative
uncertainty of the ice volume $\Sigma_{V_i}$, whereas by applying Eq. (15), $\Sigma_{S_w}$ is 0.84 times the relative uncertainty of the water
volume $\Sigma_{V_w}$, considering the values used in the original paper of Colbeck (1978). Nevertheless, it is essential to note that the
relative uncertainty produced by the melting calorimeter on the estimation of the ice volume $\Sigma_{V_i}$ is one order of magnitude
lower than the relative uncertainty produced by the freezing calorimeter on the water volume $\Sigma_{V_w}$. This renders the final
uncertainty of the two methods comparable, as shown in Fig. 2. All the mathematical details are reported in Appendix B and
relative codes. Therefore, while Colbeck (1978) argued that melting calorimetry is "inherently inaccurate", our mathematical
refinement, occurring 45 years after Colbeck's original paper, fundamentally alters the prevailing understanding that the melting
calorimeter, an indirect measurement of liquid water content, is not inherently inaccurate.

Finally, Figure 2 shows an opposite trend compared to Figure 3 in Kawashima et al. (1998). In our study, the uncertainty
$\sigma_{\theta_w}$ increases with increasing $\theta_w$. This is because we compared snow samples with the same volume and ice content but
varied liquid water content, mimicking an introduction of liquid water in the snow filling the voids. Consequently, as the liquid
water content increases, so does the snow density. In contrast, Kawashima et al. (1998) does not account for volume in their
formulation, instead adapting the snow mass to the water mass based on the parameter $R = \frac{M_w}{M_s}$. While both approaches are
mathematically correct, adjusting the snow density according to the liquid water content aligns more closely with the practical
reality of measuring liquid water content in the snowpack (Picard et al., 2022).

## 3.2 Minimizing the Instrumental Uncertainty of the Melting Calorimetry

Leveraging the framework established in Section 3.1, we can now analyze how the uncertainties in variables measurement
propagate to the final water content uncertainty $\sigma_{\theta_w^M}$ through the melting calorimetry formula (Eq. 3). Often overlooked,
uncertainty propagation analysis is a critical step for new instruments. It helps us to identify the optimal operational range for
achieving the least uncertain measurements. Given the substantial differences between our approach (volumetric formulation
with compensation for the calorimetric constant) and previous methods, a new analysis is necessary. As described in Section 2,
some of the variables involved in the calorimetry are free to be selected during the experiment. These are $V_s$, $M_s$, $M_w$, $T_w$,
and $E$ (see Eq. (3)). However, all these variables are strictly connected to each other with some implications that it is better to
account for.

Firstly, by increasing the volume sample $V_s$, we expect that the uncertainty is decreasing being $V_s$ at the denominator for all
the terms in Eq. (8). Nonetheless, the volume of the sample is linked to its mass, which increases by increasing the volume, this
has an effect of compensation on $\sigma_w^M$. This outcome paves the way for dedicated investigations into employing various sample
volumes for characterizing liquid water transport within the snowpack. Although such research lies outside the scope of this
paper, we provide here some ideas that explain the volume selected in the context of this paper. As discussed in Section 3.1, $V_s$
is generally constrained by the snow sampler used, which is normally fixed e.g., the Taylor–LaChapelle density cutter. Similarly
with the measuring of the density (Proksch et al., 2016), $V_s$ is selected focusing primarily on the resolution required to describe
$\theta_w$ in the snowpack: a small snow cutter allows to sample the small difference in $\theta_w$ within the snowpack, whereas a large



volume density sample provides snowpack bulk information. Given the high heterogeneity of $\theta_w$ (Techel and Pielmeier, 2011), one can think of sampling the snowpack with a high vertical and horizontal resolution. However, this requires performing a

large number of calorimetric analyses, hence it is not possible to assume that $\theta_w$ did not change for that time - to avoid this, in 1967 Yosida employed a parallel team of students that performed 120 measurements to describe the temporal evolution of a 160 cm snowpack over 7 hours highlighting that an average of 20 mins are required for each measurement. On the contrary, taking a snow sample that is big poses some challenges for both the selection of the calorimeter, which generally features a small aperture (see Fig. 1) and the possibility of representing specific cases such as the situations of water saturation on top

of ice layers. In practice, by considering a calorimeter with a capacity of 0.5L, $V_s$ should fall between 100 and 300 $\text{cm}^3$. For melting snow, this corresponds to a $M_s$ that ranges approximately between 60 and 150g. Within these values the influence of $V_s$ on the overall uncertainty $\sigma_w^M$ is limited. For samples of this volume, it is advisable to prioritize a smaller vertical footprint—such as a cylindrical sampler with a diameter of 4 cm and a depth of 18 cm—over a shallower penetration depth with a larger vertical footprint. This strategy aids in identifying saturated layers and ensures smoother insertion of the snow

sample into the calorimeter i.e., the diameter of the sampler is smaller than the opening of the calorimeter.

At this point, the remaining variables are $M_w$, $T_w$, and $E$. It is then mathematically convenient to introduce a new variable as done in (Kawashima et al., 1998) i.e.,

$$R = \frac{M_w}{M_s} \tag{22}$$

This allows us to plot the uncertainty as a function of the water temperature and the ratio between the masses for different

levels of $\theta_w$ (see Fig. 3). Considering a $V_s = 200 cm^3$, a dry snow density of $366 \text{kgm}^{-3}$ and a calorimeter with $E = 6.58$ g, it is possible to show that low values of R i.e. a same mass of hot water and snow and high water temperature $T_w$ are the ones that produce the best results in terms of uncertainty. However, high $T_w$ means a large temperature loss when the calorimeter is open to insert the snow sample. To minimize this loss, which otherwise has to be considered in the calorimetric formulation, a $T_w$ of 40 to 50 °C Celsius together with a quick insertion of the snow sample in the calorimeter, is a good trade-off for all the

possible cases as indicated by Kawashima et al. (1998). On the other hand it is worth stressing the fact that, even though low values of $R$ produce the best results, this has two practical implications that should not be neglected:

i) if an immersion thermometer is used (as the one represented in Fig. 1) particular attention should be devoted to the fact that the probe is properly immersed inside the water-snow mixture. Otherwise the temperature measure will oscillate. Therefore, especially if the calorimeter is very tall and the snow sampler is small and cannot be increased, $M_w$ should be increased. This

may allow a proper immersion of the temperature probe inside the snow-water mixture.

ii) If the snow sample is too big with respect to the water mass, or the water temperature is too low, the heat exchange cannot be completed i.e., $T_f < 0°C$. As recommendation we advise to keep $T_f > 5°C$. A value of $R = 2$ is a good trade-off for all the possible real cases (see Fig. 3).

Finally it is important to mention that different values of $E$ imply a change in the uncertainty (see Eq. (9)-(14)). However,

we stress the fact that this change is limited and therefore different sizes, shapes and qualities of calorimeter can potentially be



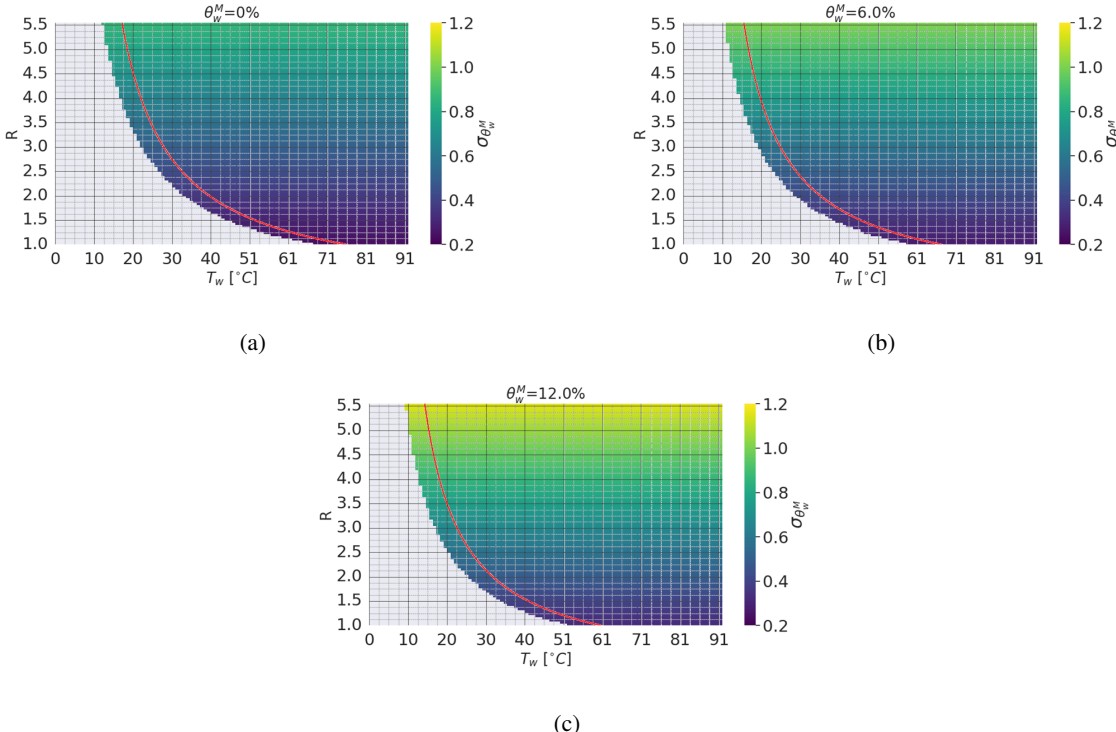

**Figure 3.** In the figure the uncertainty associated to $\theta_w^M$ is presented as a function of the experiment parameter $R$ and $T_w$. In detail, the uncertainty is presented for 3 representative scenarios of $\theta_w^M$ i.e., $0\%$, $6\%$, and $12\%$. In (a) the dry snow scenario with $\theta_w^M = 0\%$ is presented; in (b) the medium wetness scenario with $\theta_w^M = 6\%$ is presented and in (c) the limit wetness scenario with $\theta_w^M = 12\%$ is presented. Gray areas represent the cases where the calorimetric experiment results in a final temperature lower than $0°$C. The indicated red boundary signifies a "safety" threshold constrained by a final temperature of $5°$C, i.e. the points laying on the red line result in a $T_f = 5°$C. When working on the left side of the curve, we are pushing the limits of experimental feasibility. Therefore, it is advisable to exercise caution and strive to remain on the right side of this boundary. In all cases, high values of $R$ are associated with high uncertainty.

employed. Differently, a wrong estimation of the calorimeter constant can have a big impact on the accuracy of the measure of $\theta_w$ (see next section).

### 3.3 Sensitivity Analysis of the Melting Calorimetry

By studying the so-called sensitivity coefficients i.e., Eqs. (9)-(14), we can derive important information about the sensitivity

of melting calorimetry to variation of the input variables $V_s$, $M_s$, $M_w$, $T_w$, and $E$ to the final value of $\theta_w^M$. In fig.4 the variation in $\theta_w^M$ associated with the change of the experiment parameters is reported for a realistic case where the snow density is $416\text{kgm}^{-3}$, $M_s = 83.4$g, $M_w = 166.8$g, $T_f = 4.8°$C, $T_w = 40°$C, $E = 6.58$g and $\theta_w^M = 5.0\%$. Note how $R$ and $T_w$ are



derived from the considerations discussed in Section 3.2. Fig.4 allows an operator to become more aware of the impact of potential variations when taking new measurements.

Figure 4a illustrates that the error in volume measurement is directly proportional to $\theta_w$. However, the impact of this error is relatively limited. For instance, in the considered scenario where a snow box cutter with dimensions of $4 \times 2.75 \times 18\mathrm{cm} = 198\mathrm{cm}^3$ is employed, and a portion of $20\mathrm{cm}^3$ is lost due to incomplete filling i.e., the snow cutter is not filled to a depth of $2\mathrm{cm}$, the resulting $\theta_w$ measurement is overestimated by a mere $0.10\%$. This suggests the feasibility of utilizing snow samplers with variable depth, such as samplers with a moving piston, to adapt the sample size according to $\theta_w$ distribution in the snowpack.

In Fig. 4b, one can observe how even a small error of a few grams in the measurement of hot water mass can significantly impact the resulting $\theta_w$, causing a nearly 1 percentage point difference. This highlights the crucial significance of ensuring a stable, level, and well-prepared position for the scale, protected by the influence of wind gusts by a well dug snow pit and an additional shield. In the end the plate of the balance should be sufficiently large compared to the calorimeter.

The accurate measurement of the hot water temperature is equally important (see Fig. 4c). An overestimation of $1°\mathrm{C}$ in $T_w$
results in an approximate $1\%$ underestimation of $\theta_w^M$. Additionally, it is crucial to emphasize the importance of measuring $T_w$ only after the entire calorimeter reaches a stable temperature. Using a well-insulated container and promptly inserting the snow sample into the calorimeter, so that the hot water is not cooling down, are vital factors to minimize errors in the measurement of $T_w$ as seen in Section 3.4. The specification regarding the depth of immersion for the thermometer probe must be satisfied.

The misreading of $T_f$ leads to a directly proportional error in the final $\theta_w$ measurement (see Fig. 4d). Achieving an accurate
measurement of $T_f$ necessitates ensuring that the heat exchange process is fully completed. To facilitate this, it is recommended to gently shake the calorimeter, and the completion of the process can be verified by observing a clear stabilization in the temperature reading. This is usually occurring within 30 seconds after inserting the snow sample into the calorimeter. Even under harsh conditions, heat loss from the closed calorimeter remains minimal. The negligible temperature loss holds true even when the calorimeter is exposed to sunlight or experiences internal-external temperature differences exceeding $40°\mathrm{C}$. These
conditions far surpass typical operating environmental conditions.

An error in the mass measurement of the snow sample $M_s$ introduces a directly proportional error in $\theta_w^M$ (see Fig. 4e). To ensure accurate measurements, the same precautions as those taken for the measurements of $M_w$ should be followed.

Finally, it is crucial to note that an error in $E$ is inversely proportional to $\theta_w$ (see Fig. 4f). Neglecting the heat exchange with the calorimeter wall, i.e., assuming $E = 0\mathrm{g}$, results in significant errors in the final measurement of $\theta_w^M$. Therefore, when using
a new calorimeter, proper time and effort should be dedicated to accurately estimating $E$, as described in Section 3.1.

The selection of the values will be summarized in section 4 in form of a protocol that all the researchers and practitioners can follow during field experiments.

## 3.4   Random uncertainty introduced by environmental factors and operator variations

After assessing the instrumental uncertainty of the melting calorimeter, it is crucial to consider the additional error sources
that arise during the practical implementation of the experiment. These uncertainty sources primarily stem from the operator handling the various steps involved in the experiment, and the environmental factors at the measurement site. Quantifying these



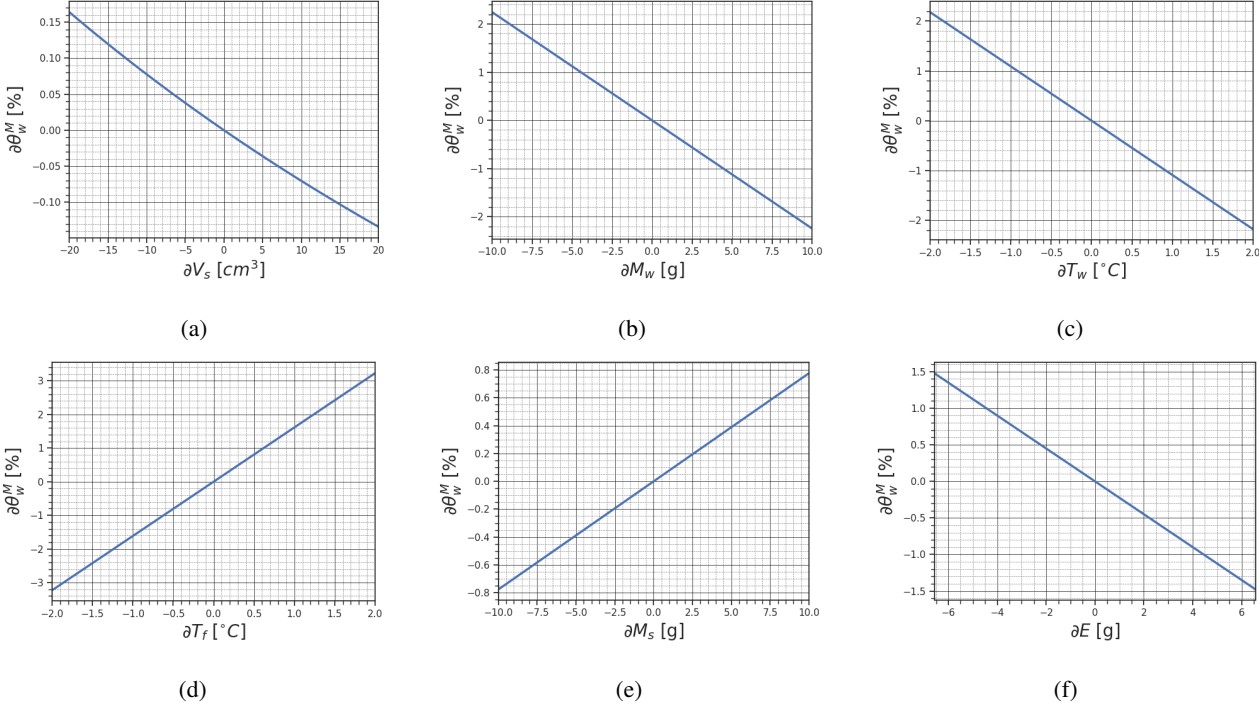

**Figure 4.** In the figure the variation of $\theta_w^M$ with respect to the six parameters that define the experiment is reported. In detail, we report in (a) the variation of $\theta_w^M$ with respect to $V_s$, in (b) the variation of $\theta_w^M$ with respect to $M_w$, in (c) the variation of $\theta_w^M$ with respect to $T_w$, in (d) the variation of $\theta_w^M$ with respect to $T_f$, in (e) the variation of $\theta_w^M$ with respect to $M_s$

and in (e) the variation of $\theta_w^M$ with respect to $E$.

errors is challenging, but it is essential to establish a robust measurement protocol to minimize their impact. It is therefore important to dig into a comprehensive analysis of these two distinct sources of uncertainty.

Field campaigns often expose researchers or practitioners to challenging environmental conditions, encompassing factors
such as wind, snowfall, rain, or high solar radiation. The temperature spectrum can fluctuate significantly, spanning from sub-zero to positive values, thereby amplifying the potential for heat loss from the hot water and inducing changes in the sample phase, respectively. Additionally, the involvement of different operators, each potentially employing slightly varied techniques due to the different interpretations, introduces a layer of variability throughout the measurement procedure.

Experiments with excessively harsh environmental conditions and/or those that deviate from established snow pit measure-
ment practices will yield unreliable results and hinder uncertainty estimation, compromising the validity of the measurement. Wind impacting the scale and direct solar radiation exposure of the snow sample are prime examples of such conditions. To quantify the impact of wind on measurement uncertainty, we conducted an experiment where a controlled breeze (around 15 km/h) generated by a fan was directed at the scale. This resulted in a tenfold increase in scale uncertainty, from 0.1g to 1.5g. By applying $\sigma_{M_w} = 1.5\text{g}$ and $\sigma_{M_s} = 1.5\text{g}$ to Eq. 8, a significant uncertainty of more than 1% will be associated to





the measurement of liquid water content. These results highlight that wind is a major factor that can invalidate calorimetric measurements by introducing excessive uncertainty. Similar to wind, solar radiation and sample handling can significantly impact $\theta_w$. Rough handling, and prolonged air and sun exposure can all alter $\theta_w$ of the snow sample. Fortunately, following basic snow pit practices like shielding the scale from wind and protecting the sample and instruments from direct sunlight can prevent measurement failure and allow a correct computation of the associated uncertainty. In most cases, this allows us to

assume $\sigma_{M_w}$ and $\sigma_{M_s}$ equal to the scale uncertainty.

Therefore, by assuming proper adherence to common field protocols, the main factors that contribute to random uncertainties include: i) prolonged exposure of the sample to air temperature, producing liquid water; ii) variations in the time the calorimeter insulated cup is open for sample insertion, affecting heat exchange; iii) proper mixing and operator judgment of thermal equilibrium, since the snow samples immerse in the snow can find local equilibrium points before the complete melting. All

of these uncertainties can be quantified by performing many controlled experiments under varying conditions. While using reference samples with a known $\theta_w$ would be ideal, preparing such samples artificially introduces significant uncertainties. The methods employed by Kawashima et al. (1998) and by Fasani et al. (2023) involve introducing liquid water into dry snow, which is a complex operation. A critical challenge in this approach is maintaining precise temperature control. Both the snow sample, the air, and the added liquid water need to be at exactly $0°C$ for successful preparation. Deviations from this ideal

temperature can significantly alter the effective $\theta_w$ of the reference sample. Given the inherent difficulty in achieving such precise temperature control, we recommend avoiding this approach.

To address the challenges associated with reference samples, we designed two alternative experiments. In the first experiment, we used dry snow samples and measured the final system temperature. This experiment serves as a proxy for evaluating the impact of sample handling by the skilled operator on measurement uncertainty. In the second experiment, we investigated

the combined effects of environmental conditions and operator variability. Here, we controlled the air temperature and recruited a group of volunteers with varying levels of snow field experience. To assess uncertainties stemming from wet snow reference sample preparation in a controlled manner, we employed ice cubes. The significant temperature difference between the ice cubes and the room temperature allowed us to simulate worst-case scenarios. This approach provides a valuable assessment of uncertainties without introducing the complexities associated with artificial wet snow samples. Moreover, these two

experiments allow us to appreciate the effectiveness of the melting calorimeter.

### 3.4.1 Dry snow experiment

As done for Eq. (1), we can derive the expression for the energy budget in case of dry snow:

$$\underbrace{Q_{hot\ water}}_{M_w C(T_f - T_w)} + \underbrace{Q_{calorimeter}}_{M_{cal} C_{cal}(T_f - T_w)} + \underbrace{Q_{warming\ snow}}_{M_{dry\_snow} C_{ice}(T_f - T_{dry\_snow})} + \underbrace{Q_{snow\ melting}}_{L M_{dry\_snow}} - \underbrace{Q_{melted\ snow}}_{M_{dry\_snow} C(T_f - T_{mi})} = 0 \qquad (23)$$

From eq. 23 can be retrieved the final temperature of the system as:




$$T_f = \frac{C_i M_{dry\_snow} T_{dry\_snow} - L M_{dry\_snow} + C(M_w + E)T_w}{C(M_w + E) + M_{dry\_snow} C} \qquad (24)$$

The associated instrumental uncertainty can be derived as follows (IEC et al., 1993):

$$\sigma_{T_f}^{\text{Ins}} = \sqrt{\left(\frac{\partial T_f}{\partial M_s}\right)^2 \sigma_{M_s}^2 + \left(\frac{\partial T_f}{\partial T_s}\right)^2 \sigma_{T_s}^2 + \left(\frac{\partial T_f}{\partial M_w}\right)^2 \sigma_{M_w}^2 + \left(\frac{\partial T_f}{\partial T_w}\right)^2 \sigma_{T_w}^2 + \left(\frac{\partial T_f}{\partial E}\right)^2 \sigma_E^2} \qquad (25)$$

Where $T_s$ and $M_s$ are respectively the temperature and the mass of the dry snow sample.

The dry snow experiment revealed a lower overall uncertainty than the inherent instrumental uncertainty (refer to Fig. 5a). The results demonstrate that well-trained operators can significantly minimize uncertainties associated with sample handling and lid opening during calorimetric analysis. Since for this experiment we always obtained an experimental uncertainty lower than the instrumental uncertainty, we made only a total of 5 experiments. We instead focus on the heat lost when the calorimeter is opened for snow sample insertion. We observed a temperature loss of $0.2°$C after 10 seconds with a $40°$C difference between the air and the water temperature. Since sample insertion itself takes a maximum of 2 seconds by a skilled operator, the heat lost is negligible. This is another evidence that the instrumental uncertainty found in Section 3.1 can be applied in this case. This last finding aligns with the observations from Kawashima et al. (1998) (see Section titled "the effect of the heat loss by opening the lid" of the original paper). The results of Fig. 5a further emphasize the effectiveness of the melting calorimeter design for accurate snowmelt measurements, especially when combined with proper operator training and handling techniques.

### 3.4.2 Ice cube experiment

The second experiment aims at quantifying the impact of the operator variability at different air temperatures. To this end, we recruited a group of volunteers to perform a large number $N$ of calorimetric experiments i.e., $N > 30$ following basic guidelines. To simulate variations in air temperature and operator performance, we assemble a diverse group of individuals with varying expertise in the field of snow measurements. These participants are tasked with conducting measurements within an environment spanning temperatures from -5 to 20 °C. This multifaceted approach aims to capture the intricate interplay between operator influence and temperature differentials. The experimental setup, resembling the dry snow calorimetric experiment, aims to compare the measured final temperature $T_f^{\text{Exp}}$ of the mixture of hot water after the addition of the ice cube with the theoretical temperature $T_f$ calculated using the calorimetric formula. Specifically, the experiment involved a predetermined mass $M_{ice\_cube}$ of ice sample at a known temperature $T_{ice\_cube}$ and a mass $M_w$ of water at a known temperature $T_w$. The ratio between the ice mass $M_{ice\_cube}$ and the water mass $M_w$ was kept constant to $\frac{1}{10}$. The temperature of the ice cubes was determined by leaving the cubes for at least 24 hours in a refrigerator with monitored temperature. As done in Eq. (1), we can express the experiment as an energy budget by equalizing the energy carried by the hot water ($Q_{hot\ water}$) and the internal wall of the calorimetry, in thermal equilibrium with the hot water ($Q_{calorimeter}$), with the sinks consisting of the ice cube ($Q_{ice\ melting}$) and the water derived by melted ice ($Q_{melted\ ice}$). Starting from the energy balance presented in Eq. (23) the




final temperature $T_f$ can be derived from Eq. (24) by changing the mass and temperature of the dry snow with the mass and

the temperature of the ice cube.

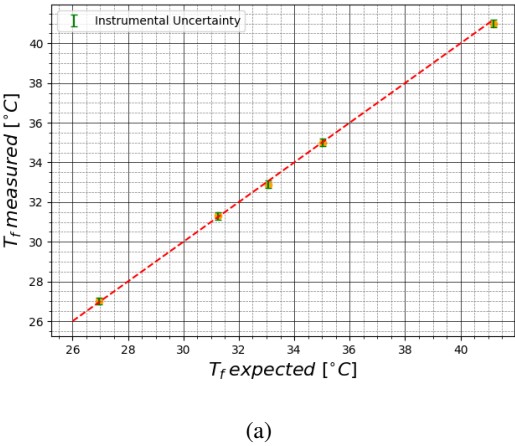 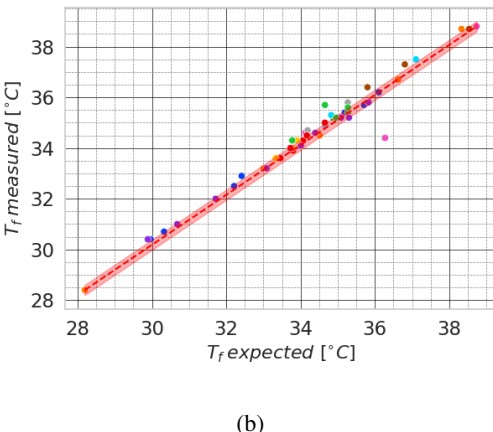

(a)                                                        (b)

**Figure 5.** Results of the calorimetric experiment performed using (a) dry snow; and (b) ice cubes by a group of volunteers. The measured final temperature (y-axis) is compared with the theoretical final temperature (x-axis) derived from Eq. (24). In (a) the green buffer represents the instrumental uncertainty of all the measurements using Eq. (25), which is slightly visible to its low values. In (b) the different colors of the dots represent the different people who performed the experiment. The red area represents the instrumental uncertainty of all the measurements using Eq. (27).

The outcome of the experiment is presented in Fig. 5b. From a qualitative point of view, the figure demonstrates a good agreement between the measured and theoretical values indicating the effectiveness of the melting calorimetry. The small spread of the data indicates that the experiment is largely reproducible among different operators, even though some operators generate larger errors. The missing of a bias indicates that there are no large systematic errors. From a quantitative point of

view, the experimental uncertainty can be calculated from this data as the Root Mean Square (RMS) of the differences between the measured final temperature $T_f^{\text{Exp}}$ and the theoretical final temperature $T_f$ as follows (IEC et al., 1993).

$$\sigma_{T_f}^{\text{Exp}} = \sqrt{\frac{\sum_i^N (T_{f_i} - T_{f_i}^{\text{Exp}})^2}{N}} \tag{26}$$

This results in $\sigma_{T_f}^{\text{Exp}} = 0.5$. However, in our setup, the thermometer uncertainty of $0.1^\circ$C and the scale uncertainty of $0.1$g limited our ability to definitively isolate the impact of environmental and operator variability on the measured liquid water

content. A t-test for null hypothesis verification (Rouder et al., 2009) applied to Eq. (26) sometimes rejected the null hypothesis i.e., the small resolution of our instruments resulted in statistically insignificant differences between the estimated and measured temperatures. To more effectively characterize the uncertainty arising from environmental factors and operator variability, this experiment should be repeated with a higher precision thermometer (e.g., $\sigma_T = 0.01^\circ$C), which was not available during the experiment.



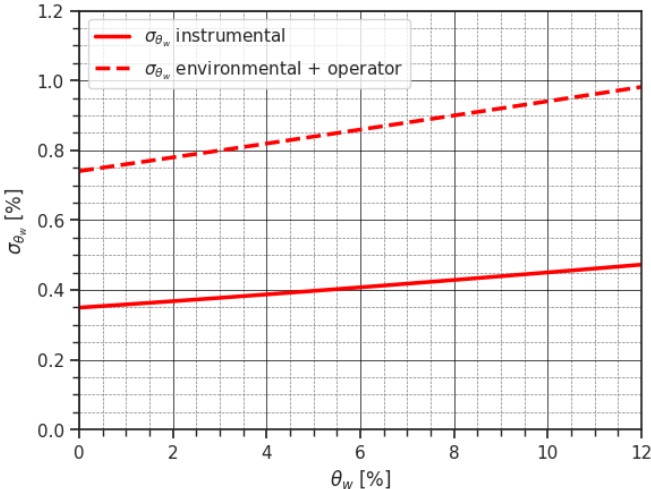

**Figure 6.** Comparison of the propagation of uncertainty considering the instrumental uncertainty (solid red line) and random uncertainty introduced by variations in operators and environmental factors (dashed red line) for the melting calorimeter.

Despite the limitations of our current instrumentation, let us explore a hypothetical scenario where the experimental uncertainty $\sigma_{T_f}^{\text{Exp}} = 0.5$. This total uncertainty can be further broken down as the sum of the instrumental uncertainty $\sigma_{T_f}^{\text{Ins}}$, the operator induced uncertainty $\sigma_{T_f}^{\text{Ope}}$ and the environment induced uncertainty $\sigma_{T_f}^{\text{Env}}$. The instrumental uncertainty $\sigma_{T_f}^{\text{Ins}}$ can be calculated as follows (IEC et al., 1993):

$$\sigma_{T_f}^{\text{Ins}} = \sqrt{\left(\frac{\partial T_f}{\partial M_i}\right)^2 \sigma_{M_i}^2 + \left(\frac{\partial T_f}{\partial T_i}\right)^2 \sigma_{T_i}^2 + \left(\frac{\partial T_f}{\partial M_w}\right)^2 \sigma_{M_w}^2 + \left(\frac{\partial T_f}{\partial T_w}\right)^2 \sigma_{T_w}^2 + \left(\frac{\partial T_f}{\partial E}\right)^2 \sigma_{E}^2} \tag{27}$$

At this point, the uncertainty associated to the operator variations $\sigma_{T_f}^{\text{Ope}}$ and the environmental factors $\sigma_{T_f}^{\text{Env}}$ can be retrieved as a simple difference i.e., $\sigma_{T_f}^{\text{Ope}} + \sigma_{T_f}^{\text{Env}} = 0.3°\text{C}$. Given the fact that the operational steps of this experiment are exactly the same of the operational steps for the melting calorimeter, we can use the derived $\sigma_{T_f}^{\text{Ope+Env}}$ to update the uncertainty calculated in Section 3.1 by adding this uncertainty to the final temperature measurement uncertainty $\sigma_{T_f}$. In Fig. 6, we observe how this new characterization of the uncertainty affects the uncertainty in the estimation of $\theta_w$ with respect to only considering
the instrumental. While the limitations of our current instrumentation prevent definitive conclusions from this analysis (i.e., non-statistically significant results), this exercise demonstrates the value of a quantitative approach to understanding complex uncertainty sources. This framework can be applied in future studies with more precise equipment to accurately characterize these uncertainties. A similar analysis could be also applied to the freezing calorimeter, considering the losses associated with the measurement time required by this technique. However, this is beyond the scope of the paper.





## 4 Melting Calorimeter Protocol

In this section, we will provide a comprehensive summary of the analyses conducted throughout the paper and propose a practical measurement protocol to be followed during field campaigns. First, the necessary materials need to be prepared, including:

**Calorimeter** This could be a commercially available insulated container designed to maintain the temperature of beverages or food. The calorimeter constant must be known or derived for accurate measurements (e.g., with Eq. (2)). In this work we used a commercial insulated container i.e., Stanley Classic Legendary Food Jar made of stainless steel (i.e., SAE 304) $C_{\mathrm{cal}} = 500 \, \mathrm{Jkg}^{-1}\mathrm{K}^{-1}$ and $M_{\mathrm{cal}} = 69.1$ g, $E = 6.58$ g. The uncertainty of this measurement is only related to the uncertainty of the scale used to weigh the container i.e., 0.1g. These values have been measured, and they are not provided by the producer.

**High-Precision Immersion Thermometer** Used to monitor temperature changes during the experiment (we used a Hanna Instruments HI98501).

**Precision Balance** Utilized for measuring the mass of hot water and snow samples. The scale should be at least splash proof (IPX4). We used a Kern PCB-10000-1

**Supportive Surface** A level, hard and supportive surface such as Plexiglass.

**Wind Shield** A well dug shelter or an external barrier, such as a plexiglas container can be used to shield the scale from wind.

**Insulated bottle** The bottle acts as a reservoir of hot water, ensuring a continuous supply for the experiment.

**Snow sampler** This can be any of the available snow samplers for density measurements. The shape and size of the sampler must be compatible with the shape and size of the calorimeter.

The snow pit should be dug so that the snow wall is shaded from the sun. Once the snow pit is prepared, a shaded area for the scale should be established (see Fig. 7a), protecting it from solar radiation and wind. If shade is unavailable for the scale, measurements should be avoided. Alternatively, a shading protection system like the one shown in Fig. 7c can be used. Another hole should be dug for storing the snow sampler and auxiliary tools needed for sample preparation to maintain them at low temperatures. The outer surface of the pit profile should be smoothed, and the bottom should be level to ensure accurate measurement of the snow height at which the $\theta_w^M$ measurement will be taken.

Following these preparations, the following steps should be followed:

1. Warm up the water at the temperature of 40 to $50°\mathrm{C}$ and store it in the insulated bottle;

2. Tare the scale with the calorimeter and the lid with the thermometer on top;





3. Prepare the hot water inside the calorimeter in a quantity that meets the minimum immersion requirement of the thermometer and is approximately two times the sample mass (annotate it as $M_w$); If a volume of $200\text{cm}^3$ is used, approximately 200g of hot water should be used;

4. Close the calorimeter and wait for the temperature to stabilize;

5. Record the temperature $T_w$. It should be around $40°\text{C}$;

6. Tare the scale with the calorimeter and hot water (otherwise the uncertainty of the two measurements must be propagated through Eq.(7)).

7. Retrieve the snow sampler from the shaded hole and collect a snow sample from the designated height, ensuring no phase changes occur (i.e. take the sample on shade) or any loss;

8. Open the calorimeter and place the snow sample inside. Minimize the time for this step;

9. Gently shake the calorimeter;

10. Weigh the snow mass $M_s$ by placing the calorimeter on the tared scale waiting for the temperature to stabilize;

11. Once the temperature stabilizes, approximately 30 seconds - 1 minute after sample insertion, read the temperature $T_f$.

12. Empty and dry the calorimeter for subsequent measurements.

By following this protocol meticulously, the measurement of $\theta_w^M$ in the snowpack can be conducted with the accuracy and the uncertainty shown in this paper. By using the shared code provided with the paper $\theta_w^M$ and the relative uncertainty can be derived starting from the measured data.





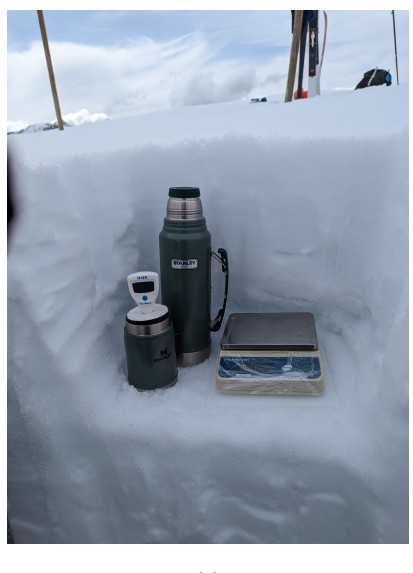 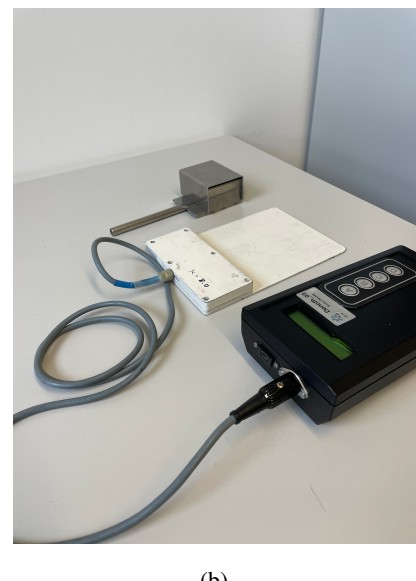 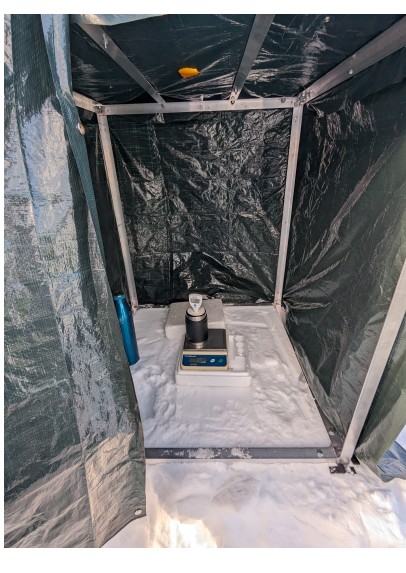

|  (a)  |  (b)  |  (c)  |

**Figure 7.** In (a) is reported the essential materials for the melting calorimeter experiment. From left to right: the calorimeter, its thermometer, the thermos storing the hot water used in the experiment, and the scale for measuring mass placed on a stable plexiglass panel. In (b) is reported from left to right, the snow cutter, used for sampling the snowpack at the Weissfluhjoch site and the Denomether used during the measurement campaign. In (c) it is possible to see the measurement configuration in case it is not possible to dig a snow pit deep enough to protect the calorimeter from wind and solar radiation. However, Measurements under extreme weather conditions are not recommended.

## 5  Experimental Application of the Melting Calorimeter protocol: $\theta_w^M$ Profiles in the Field

To assess the performance of the calorimeter in field conditions and refine the measurement protocol, a series of field activities were conducted. These activities aimed to test the accuracy and reliability of the calorimeter protocol in real snowpack and with real conditions. The field tests were carried out in Val Senales (Italy) (see Fig. 8b-d) and at the Weissfluhjoch (Swiss) (see Fig. 8c-e) with different melting snowpacks, including different snow densities and liquid water content levels spanning the period from April to June 2023. To assess the validity of the calorimetric measurements of $\theta_w^M$, additional measurements of density, specific surface area (SSA), stratigraphy, infrared imaging (IR), and $\theta_w^M$ conducted with the Denothmeter (Denoth and Foglar (1986)) were acquired. The large number of measurements allowed us to fine-tune the operations in the field according to the protocol and minimize the disturbance caused during sample collection allowing us to compare the different measurements.

### 5.1  Weissfluhjoch

The field tests performed in Switzerland were carried out at the field site of Weissfluhjoch (2536 m a.s.l.) in the area of Davos (Graubünden). In addition to being a high-altitude research station for which there is one of the longest observed time series in the world (Marty and Meister, 2012). It lies on a wide flat area partially sheltered from wind, the research field is fenced to



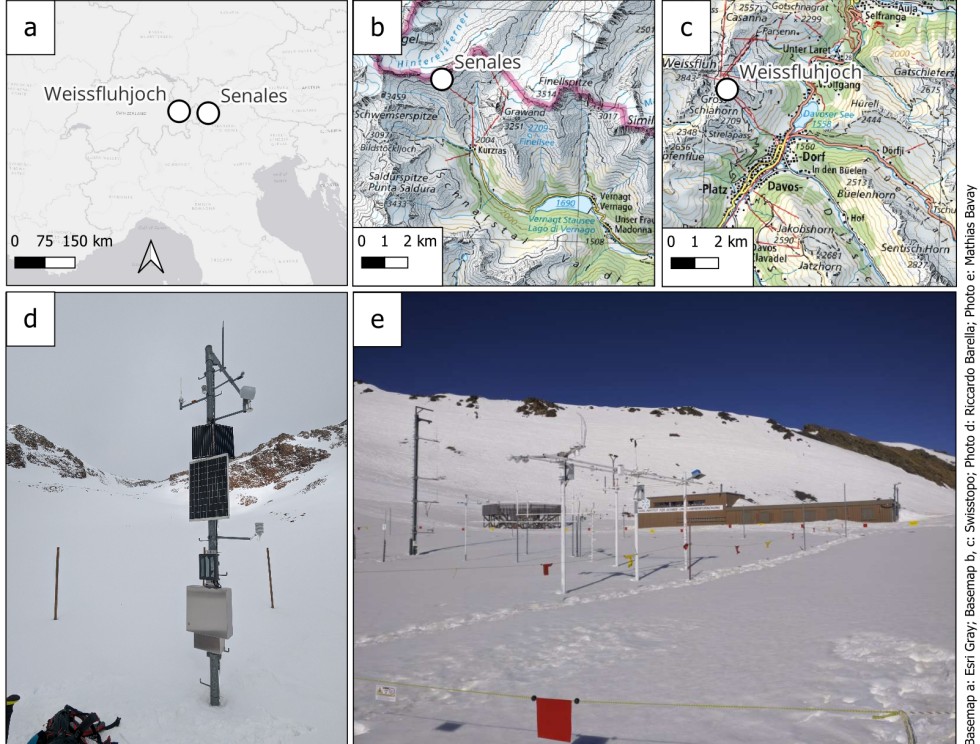

**Figure 8.** The figure gives an overview of the two sites used for testing the calorimeter. In a) the two sites can be visualized in the Alpine context, while in b) and c) the Weissfluhjoch and Senales sites can be respectively visualized in the local context. In the end in d) is shown a picture of the Senales site and in e) is shown a picture of the Weissfluhjoch site.

safeguard the snow surface as much as possible from outside intrusion and disturbance, and finally, two huts are made available to the operator, providing shelter and tools, as well as electricity, heating and internet connection. These features make the
research field particularly suitable for performing high-level measurement campaigns for various types of snowpacks.

$\theta_w^M$ measurements carried out at the Weissfluhjoch field site are part of a comprehensive measurement campaign which took place over the course of the winter season 2022-2023. The measurement campaign started on 14 February and lasted until 16 June 2023. At the beginning of the campaign, the recorded snow height was $108\ \mathrm{cm}$; on the last day of measurements, it was $50\ \mathrm{cm}$. The maximum recorded snow height during a campaign day was $192\ \mathrm{cm}$. Measurements have been performed
for a total of 36 days of measurements. A wide set of variables was measured (Snow Water Equivalent; profiles of snow temperature, density, dielectric constant, Specific Surface Area; snow roughness) using manual, electromagnetic and remote sensing systems. The vertical resolution of the snow temperature, density, permittivity and Specific Surface Area profiles is very high: measurements were taken each $10$, $4$, $3$ and $2\ \mathrm{cm}$, respectively.

Figure 9 describes the state of the snowpack at the field site of Weissfluhjoch through some properties sampled with vertical
profiles on the day of 22 May 2023. The measurements started at 08:00 Local Time (LT) in the morning underneath an



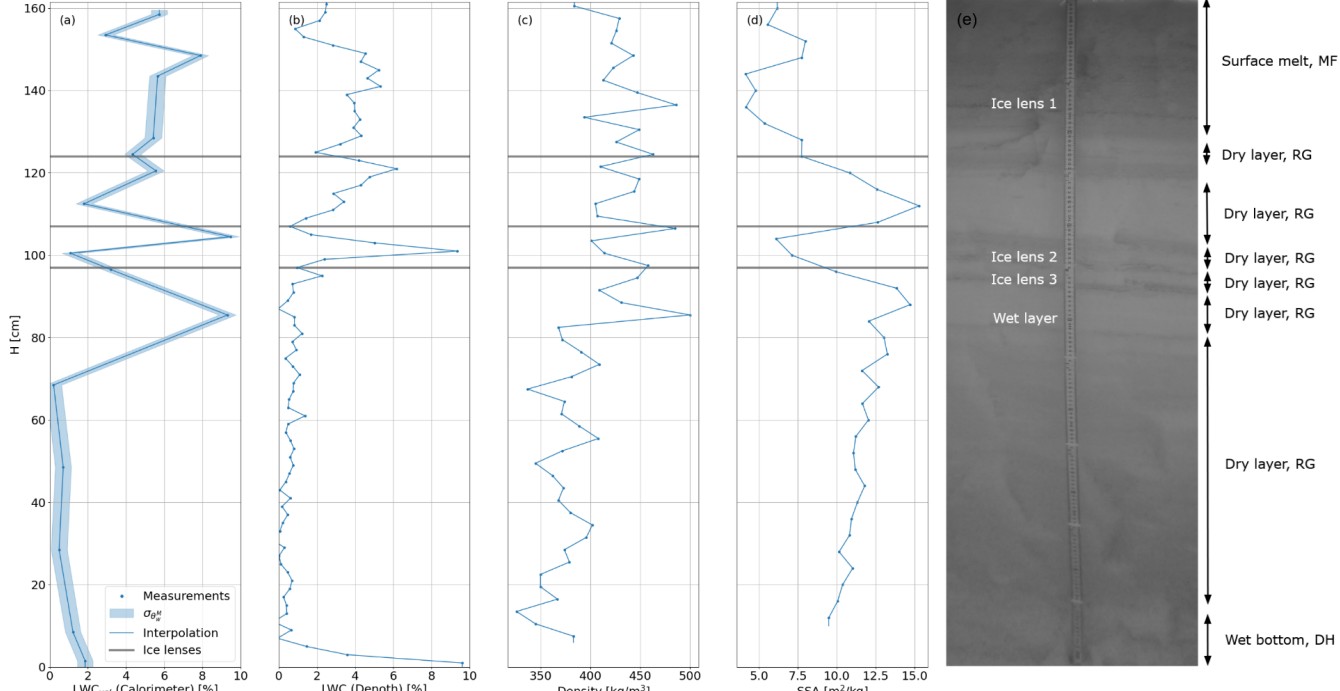

**Figure 9.** Snow profile at Weissfluhjoch on 22 May 2023. Panel (a) shows the volumetric $\theta_w^M$ profile sampled every 5 cm with the melting calorimeter and its associated uncertainty computed with Eqs. (3) and (8). Panel (b) shows $\theta_w^M$ profile sampled every 2 cm with the Denothmeter. Panel (c) shows the snow density sampled every 3 cm with the snowcutter. Panel (d) shows the Specific Surface Area profile sampled every 4 cm with the SLF InfraSnow sensor. Panel (e) is the co-registered Near Infra-Red picture on a greyscale taken on the same day before starting the measurements: different reflectance values in the picture identify differences in optical equivalent grain size. Layers are indicated according to the wetness, whereas acronyms refer to the official classification of snow on the ground (Fierz et al., 2009): MF refer to melt forms, RG to rounded grains, DH to depth hoar. On the panels, bullet points represent measurements, lines connecting them are linear interpolations, grey lines represent the position of the ice lenses as visually detected in the field.

overcast sky. The measured air temperature was $7\,°C$ and a negligible wind was blowing from North at a speed of $1\,\mathrm{km\,h^{-1}}$. The height of the snowpack was $164\,\mathrm{cm}$, the surface was smooth, and no new snow had fallen in the previous 24 hours. The temperature profile sampled every $10\,\mathrm{cm}$ from top to bottom showed that the snowpack was fully isothermal. The Near Infra-Red (NIR) photograph reported in Figure 9e illustrates quite a complex snowpack stratigraphy on that day, which along with

the stratigraphic observations helps to get a qualitative idea about general conditions and local peculiarities. It is important to notice that even though the NIR image was aligned with $\theta_w^M$ and density profiles through a rigorous co-registration procedure, ensuring that the meterstick depicted in the middle of the images aligns with the y-axis of the plots, the stratigraphic features of the snowpack and the vertical profiles shown in Fig. 9 were sampled one next to each other along an approximately 2-meters





wide wall. Given the irregularity of the layers, the location of the ice lenses recorded in the field and shown in Fig. 9 does not
match perfectly with the NIR image.

Figure 9e is annotated with qualitative comments about the state of the snowpack on the measurement day, in terms of
wetness, layers and types of grains. Surface melt was observed on the superficial layer, blocked by a layer of transformed
snow. Between H=124 cm and H=121 cm, the first ice lens was observed. Between the first ice lens and the upper transformed
layer, the snow seemed drier than on the surface. Other dry layers were observed until H=98 cm, spaced out by two more ice
lenses. Just below the last ice lens the snow seemed dry, but it was observed to become wetter towards the ground. Finally, water
runoff was found on the bottom of the snow pit. Figure 9(a) and 9(b) show $\theta_w^M$ sampled with the melting calorimeter and with
the Denoth every 5 and 2 cm, respectively; Fig. 9(c) shows the snowpack density sampled every 3 cm with the snowcutter; Fig.
9(d) shows the Specific Surface Area (SSA) sampled every 4 cm by means of the SLF InfraSnow sensor (FPGA Company).
Surface melt is captured by both the calorimeter and the Denoth, although $\theta_w^M$ on the surface measured with the calorimeter
is notably higher. The layer of transformed snow is highlighted by $\theta_w^M$ values between 4 and 5% by both the calorimeter and
the Denoth, which basically agree considering the possible natural variations. Ice lenses are characterized by drops in $\theta_w^M$
values (this is particularly well observed with the Denoth because of its higher measurement vertical resolution) and a locally
higher density. Ice lenses are observed to block the drainage of $\theta_w^M$ to the bottom: in between ice lenses the snow is drier and
characterized by a drop in $\theta_w^M$ which can be observed with both the calorimeter and the Denoth (Fig. 9(a) and 9(b)), a local
decrease in density (Fig. 9(c)) and a local increase in SSA, which increases with smaller and drier grains (Fig. 9(d)).

Figure 9(a) shows a very high $\theta_w^M$ value at 87 cm from the bottom. As can be qualitatively observed from the NIR picture
in Fig. 9(e), that height coincided with an extremely wet layer of snow. This was well captured by the calorimeter and can be
confirmed by the fact that at that height the highest snow density and SSA were recorded (Fig. 9(c) and 9(d), respectively).
Interestingly, despite the high vertical resolution of the measurements, the Denoth seemingly did not identify this wet layer.
This can also be due to a very localized point of water saturation. Finally, $\theta_w^M$ values measured by both the calorimeter and the
Denoth show that from H=87 cm to the bottom, the snowpack is releasing meltwater: from close to zero, values increase to 2
and 10% at the bottom, respectively. The release of water can also be observed as a gradual decrease of snow density below
the highest value measured at H=87 cm and, qualitatively, in a gradual decrease of SSA. This detailed analysis that compares
the stratigraphic profile, density profiles, and the liquid water content profiles reveals how these complementary data sources
provide a comprehensive picture of the snowpack conditions.

We compared the Denothmeter measurements with the melting calorimetry measurements, particularly in non-saturated
layers where we saw that the Denothmeter underestimates $\theta_w$ (Perla, 1991). Following the procedure outlined in Boyne and
Fisk (1987), we analyzed the 16 measurements where both instruments sampled the same LWC conditions for the profile of
Fig. 9 i.e., leaving out the measurement at H=87 and H=0 cm. The results showed a mean difference of $0.96\%$ and a standard
deviation of $1\%$ between the two methods (please consider that the setup for this experiment was not meant to sample in the
exact same place with the two techniques). Nonetheless, these values are mostly in agreement with the values of the original
paper by Boyne and Fisk (1987) i.e. $0.35$ and $1.13\%$ when comparing alcohol calorimeter and Denothmeter. This supports our
analysis that the melting calorimeter, if correctly used, can provide results indirectly comparable to the freezing calorimeter





and also dilution, since the main conclusion of the paper by (Fisk, 1986) is that the "four techniques - freezing and melting
calorimetry, dilution and Denothmeter - are equivalent".

Finally, it is interesting to note that applying the original formulation by Kawashima et al. (1998) implies an overestimation
of LWC by about $3.5\%$ on average compared to using the calorimetric constant, and a larger uncertainty estimation of more
than $1\%$.

## 5.2 Schnalstal

The Italian test site is located in Schnalstal, in South Tyrol. This site has been chosen because the high altitude ($\sim$3000m)
guarantees the presence of abundant snow and a long-lasting melting season, and it is well-served by lifts and roads, making it
easy to access.

The snowpack at the Schnalstal field site, with a height of 79 cm, was profiled on 7 June 2023, at 12:30 LT. During the
measurements, the air temperature stood at 1.8 °C, and there was negligible wind speed, ensuring relatively stable conditions
for the assessment. The snowpack structure was as follows: from the surface down to 65 cm, a layer of recent snow was
observed, which had undergone wetting due to temperature and solar radiation, which were relatively high that day. The
subsequent layer, spanning from 65 cm to 57 cm, contained three prominent ice lenses. Notably, a significant amount of $\theta_w$
was trapped within this layer, contributing to its distinctive characteristics.

Continuing downwards, the layer from 57 cm to the base at 0 cm was characterized by coarse snow crystals with size from
1 to 2 mm, exhibiting lower water retention capability. Importantly, there was no noticeable $\theta_w^M$ presence within this layer.
However, the lowermost 20 cm of the snowpack presented some noticeable challenges. This section displayed loose and coarse
crystals, leading to difficulties in proper sampling and ensuring optimal coupling between the Denoth instrument and the snow.
Consequently, these lower 20 cm were excluded from the subsequent analysis to ensure data accuracy and reliability.

Figure 10 (a) depicts the volumetric liquid water content ($\theta_w^M$) profile, measured with the melting calorimeter technique,
along with its corresponding uncertainty, as computed using Eqs. (3) and (8). Panel (b) of the figure portrays $\theta_w$ profile derived
from Denoth measurements. Panel (c) showcases the snow density associated with the measurement points obtained from the
calorimeter. These profiles collectively provide valuable insights into the internal structure of the snowpack and its distribution
of liquid water. Specifically, the $\theta_w$ profile reflects the typical conditions in 2023 of the European alpine snowpacks, where
ice lenses were often present. Despite a significant amount of snowfall occurring in spring, the melting process was hindered
by the presence of these ice lenses, impeding the transport of water to the ground. We can notice how the Denothmeter un-
derestimates the superficial $\theta_w$ as for WFJ. Finally, to verify the consistency of our measurements, we repeated a subset of
profile measurements at very close distances. These repeated measurements yielded very similar $\theta_w$ values inside the uncer-
tainty range, demonstrating the good stability of the melting calorimetry technique and the minimal influence of proper sample
handling.





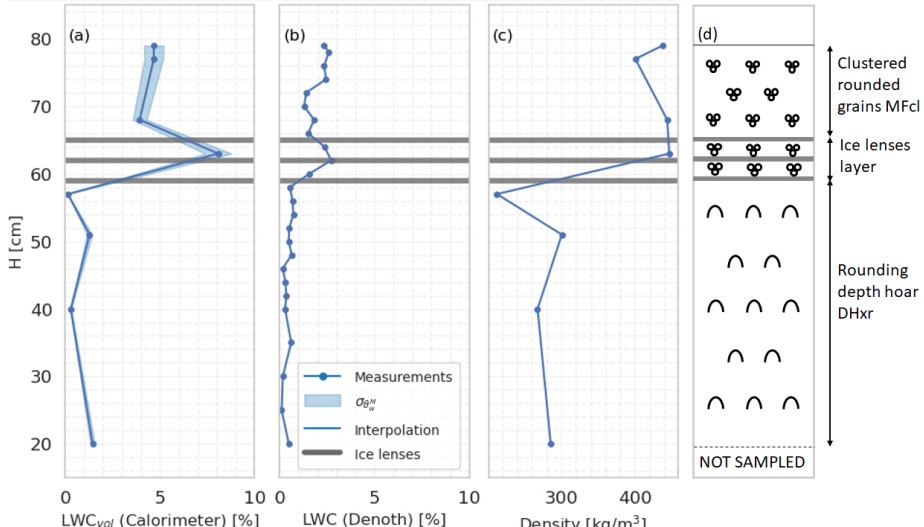

**Figure 10.** Snow profile at Schnalstal on 7 June 2023. Panel (a) displays $\theta_w^M$ profile generated using the melting calorimeter, including its associated uncertainty calculated through Eq. (3). Panel (b) exhibits the $\theta_w$ profile sampled with the Denoth. Panel (c) illustrates the snow density associated with the calorimeter measurement points. In panel (d) it is reported a scheme of the profile since for that site no IR photos were available.

## 6  Discussion

The primary objective of this study was to provide a rigorous exposition of the melting calorimetric technique for estimating liquid water content. The literature outlines two calorimetric approaches used for the estimation of $\theta_w$. Prior to our paper, a rigorous mathematical analysis by Colbeck (1978) argued against the use of melting calorimetry, a concept that echoed in subsequent works, thus becoming widely accepted. We have clarified and expanded upon Colbeck's findings, demonstrating unequivocally that the two techniques are comparable in terms of accuracy. While freezing calorimetry still holds an edge, particularly considering instrumental uncertainty alone, our analysis serves to dismiss the melting calorimeter accusations. The additional sources of uncertainties that stem from the challenging operational conditions of using a freezing calorimeter still require further investigation but might result in large uncertainty especially when used by inexperienced operators (Fisk, 1986). However, the melting calorimetry reputation for inaccuracy is partially justified by limitations in its mathematical modeling and uncertainty analysis. Building new concepts upon these unconsolidated works leads to the propagation of errors that may hinder the advancement of liquid water content measurement techniques. We therefore found it necessary to invest significant effort in thoroughly analyzing all available material.

We present the correct formulations i.e., Eq. (3) and Eq.(6) for both melting and freezing calorimeters, respectively expressed in %vol and incorporating the calorimetric constant. While the freezing formulation was established upon its introduction in literature (Jones et al., 1983), the melting formulation lacked compensation for heat exchange within the calorimeter, a discrepancy we noted since Yoshida's initial works and reiterated by Kawashima et al. (1998). This was pointed out from





the first applications of the melting calorimeter Halliday (1950). The concept of the calorimetric constant, denoted as $E$, was initially introduced for freezing calorimeters (Jones et al., 1983; Boyne and Fisk, 1987; Austin, 1990), but had not been extended to melting calorimetry until a recent study by Fasani et al. (2023). Despite the existence of the calorimetric constant
in literature, various methodologies for its estimation are available, which lead to different implications. We clarify the most effective method for estimating the calorimetric constant ($E$), considering various approaches documented in the literature and evaluating their impact on uncertainty reduction. The detailed mathematical analysis is provided in Appendix A and related Python code. Notably, the freezing formulation historically utilized by Jones et al. (1983) emerges as very uncertain, as corroborated by Austin (1990). Fasani et al. (2023) attempted the estimation of $E$ using artificial wet snow (i.e., adding a
given quantity of water to the snow sample) and inverting Eq. (3). This procedure is the most uncertain and should be avoided.

We generally doubt the effectiveness of preparing artificial samples as reference samples with a given $\theta_w$ as done in (Fasani et al., 2023; Kawashima et al., 1998), because the assumption that the added water exchange heat with the whole ice matrix cannot be always satisfied. Therefore, the selected approach for estimating $E$ in (Fasani et al., 2023), which is characterized by a significant uncertainty propagation, coupled with strong assumption and non-stringent experimental protocols such as
elevated values of $R$ (as depicted in Fig. 3 the impact of high R on measurement uncertainty is large), yields results impacted with noise. This renders the interpretation of the presented results challenging and subject to dispute. The impression is that the manner in which water is added to the snow samples, introduces a systematic bias, varying with the amount added to which a large uncertainty is added (see Fig. 5 of the original paper).

We introduce a correct formulation for estimating volume measurement uncertainty according to the GUM (IEC et al.,
1993), emphasizing its importance for facilitating comparisons between measurements, accounting not only for instrument uncertainty but also for variations in measurement setups, such as the value of R or the initial water temperature. While the error propagation method proposed by Jones et al. (1983) aligns with our approach (and leads to similar results when expressed in sum of absolute uncertainty per mass), the Kawashima et al. (1998) formulation for melting calorimeters relies on erroneous approximations. Specifically, the transition from eq. 3 to eq. 4 of the original paper lacks clarity regarding the underlying
approximation. Moreover, the assumption to maintain constant the final temperature for all the $\theta_w$ cases to simplify eq. 4 to eq. 7 (of the original paper) is invalid since the final temperature inherently depends on $\theta_w$. Consequently, Fig.3 of the original paper misrepresents the uncertainty of the melting calorimeter, even mathematically correct.

Colbeck (1978) asserts that the melting calorimeter is "inherently inaccurate", a notion echoed by the numerous subsequent studies. However, our analysis, detailed in Appendix B and provided in form of codes, demonstrates that this assertion is
not completely true. By calculating the relative uncertainties of ice and water volumes and translating them into absolute uncertainty of $\theta_w$, we refine and extend Colbeck's reasoning in section 3.1 of our paper, yielding Figure 2. This mathematical refinement, occurring 45 years after Colbeck's original paper, fundamentally alters the prevailing understanding. We now know that the melting calorimeter, an indirect measurement of $\theta_w$, is not inherently inaccurate, a significant breakthrough that we believe researchers in the field will appreciate. Moreover, our findings indicate that the performance of the melting
calorimeter is comparable to the Denothmeter under certain circumstances, which was validated against freezing calorimetric measurements. This challenges the general notion propagated by Linlor (1975) of the melting calorimeter overestimation



compared to freezing methods, which at this point should be possibly attributed to the missed calorimetric constant. This insight could be leveraged in (Webb et al., 2022), and related datasets, which at the moment show a bias similar to our WFJ measurements when the calorimetry compensation is not applied.

Leveraging our proposed formulation, we have developed and validated a field protocol for measuring liquid water content using the melting calorimeter. This formulation allows us to analytically track how uncertainties from the instrument, operators, and environment propagate to the final $\theta_w$ value. While some steps of the field protocol may resemble those in Kawashima et al. (1998), their approach lacked a correct theoretical foundation as discussed before. Our analysis provides guidance on various aspects, including calorimeter design, hot water temperature and mass relative to the snow sample, and the optimal sample

volume considering liquid water distribution within the snowpack. To our knowledge, this is the first attempt to quantify random uncertainty related to environmental and operator variability. While the limitations of the available instrument prevented a fully comprehensive uncertainty analysis, we believe that we pave the way for investigating the impact of operator variability on $\theta_w$ measurements remains an intriguing topic for future research and development.

Sample handling during extraction from the snowpack remains a key source of uncertainty. Mechanical shocks, contact with
sampler material, large temperature difference from the snow and air, and shading from solar radiation during extraction can all generate liquid water. Additionally, the impact of potentially prolonged pit exposure, a necessity for obtaining spatially detailed liquid water content profiles for deep snowpack, on water transport within the snowpack remains an open question. While our primary focus here is on the melting calorimeter accuracy for field-based $\theta_w$ measurements, and not its application in water transport studies, the results suggest that following the protocol strictly, including measures like shading the pit from
solar radiation, can mitigate this issue and provide important information about water transport inside the snowpack. Further investigation is anyway require to properly assess the influence of pit exposure.

In a field comparison at the Italian and Swiss sites, we measured liquid water content using both the melting calorimeter and the Denothmeter. By comparing our findings with results from Boyne and Fisk (1987) (which compared the freezing calorimeter, alcohol calorimeter, dilution technique, and Denothmeter), we established that our novel formulation for the melt-
ing calorimeter yields results indirectly comparable to the established freezing and dilution methods. This finding validates our approach and strengthens the case for the melting calorimeter as a reliable measurement tool. The obtained liquid water profiles of Figures 9 and 10 corroborate the observations in (Perla, 1991) that the Denothmeter provides accurate measurements for low liquid water content $\theta_w$ – typically below $5-8\%$ the uncertainty is between $0.5-1\%$ – but underestimates $\theta_w$ in very wet or soaked snow. The melting calorimeter, when employed with the proposed formulation, has the potential to expand the already
substantial data set established by (Perla, 1991). This expanded data set could be instrumental in refining the calibration of the relationship between wet snow dielectric constant and $\theta_w$, potentially leading to a clearer understanding of the physical factors that limit the unique determination of water content through dielectric measurements (Picard et al., 2022; Colbeck, 1980; Camp, 1992).

Finally, we have made available all the codes necessary for calculating $\theta_w$ and its associated uncertainty based on the
measured variables. Additionally, we offer the complete set of codes required to reproduce the results presented in our paper. This transparency ensures the reproducibility and verifiability of our findings, enabling fellow researchers to validate our





methodology and build upon our work with confidence. Crucially, with the provided methodology and codes, each calorimetric measurements can be provided alongside its associated uncertainty, which is calculated to account for variations in measurement setups, which transcend subjective interpretation. We advocate for this practice to become the standard for all future

measurements, as it ensures transparency and robustness in data interpretation. By adhering to such standards, researchers can enhance the reliability and comparability of their findings, ultimately advancing the field of $\theta_w$ estimation. We believe that sharing such resources promotes collaboration and accelerates progress in the field, avoiding misunderstanding based on subjective, incomplete, or even wrong reasoning.

## 7 Conclusions

The potential of melting calorimetry for measuring snow liquid water content has long been overshadowed by misconceptions about its accuracy.

 This paper, challenged this perception by comparing the melting and freezing calorimetry techniques, focusing on their applicability for measurements in the field. While freezing calorimetry still holds an edge, our findings indicated that the measurements obtained using the melting calorimeter are still accurate enough for a meaningful analysis of liquid water content

in the snowpack, offering at the same time notable practical advantages. To support our claims, we had thoroughly examined and propagated uncertainties, encompassing not only instrumental factors but also variations arising from the operational procedures and environmental conditions. As a result, we devised a field protocol that effectively minimizes these uncertainties. The protocol includes specific instructions on the amount of hot water to be used, its temperature, the size of the calorimeter, the masses involved, and other crucial details for controlling the uncertainty during the experiment replication. This protocol was

applied in two different test sites in Italy and Switzerland by two different research groups with different melting calorimeters. The results, compared to independent measurements of dielectric constant of the snow, showed how the application of the proposed protocol to the melting calorimeteric measurements is able to properly track the wet front penetration inside the snowpack in an accurate way.

 In conclusion, this research promotes the wider adoption of melting calorimetry as a reliable field tool for quantifying liquid

water content in snowpacks. This capability offers significant advantages for calibrating or validating new and potentially non-destructive methods based on electromagnetic interactions with wet snow.

*Code and data availability.* The code to calculate $\theta_w$ and its uncertainty from the melting calorimetry analysis will be made available along with the code for generating the figures in the paper in a dedicated GitLab of the Snowtinel project.

## Appendix A

In order to derive the calorimetric constant $E$, if information regarding the material and mass of the calorimeter cannot be obtained, the heat-balance principle can be applied. In the literature, the calorimeter constant was determined using a basic





heat-balance principle (Jones et al. (1983); Austin (1990)). When warm fluid is mixed with cold fluid in the calorimeter bottle, the heat lost by the warm fluid must be equal to the heat gained by the cold fluid and the bottle itself.

For the case of water, the heat-balance equation is given by:

$$\underbrace{Q_{hot\,water}}_{M_w C(T_f - T_w)} + \underbrace{Q_{cold\,water}}_{M_{cw} C(T_f - T_{cw})} + \underbrace{Q_{calorimeter}}_{M_{cw} C_{cal}(T_f - T_{cw})} = 0 \tag{A1}$$

Therefore we can obtain the calorimetric constant expresed in equivalent water mass as follows

$$E = \frac{M_w(T_f - T_w)}{(T_{cw} - T_f)} - M_{cw} \tag{A2}$$

In this equation, $M_w$ and $T_w$ represent the mass and temperature of the hot water, while $M_{cw}$ and $T_{cw}$ represent the mass and temperature of the cold water, respectively. By analyzing the uncertainty propagation of the measurements on $E$ with the

same approach presented in 3.1 we can retrieve the uncertainty on the estimation of E, $\sigma_E$:

$$\sigma_E = \sqrt{\left(\frac{\partial E}{\partial M_w}\right)^2 \sigma_{M_w}^2 + \left(\frac{\partial E}{\partial M_{cw}}\right)^2 \sigma_{M_{cw}}^2 + \left(\frac{\partial E}{\partial T_w}\right)^2 \sigma_{T_w}^2 + \left(\frac{\partial E}{\partial T_f}\right)^2 \sigma_{T_f}^2 + \left(\frac{\partial E}{\partial T_{cw}}\right)^2 \sigma_{T_{cw}}^2} \tag{A3}$$

The partial derivatives in Eq. (A3) are calculated as follows:

$$\frac{\partial E}{\partial M_w} = \frac{T_f - T_w}{T_{cw} - T_f} \tag{A4}$$

$$\frac{\partial E}{\partial M_{cw}} = -1 \tag{A5}$$

$$\frac{\partial E}{\partial T_w} = -\frac{M_w}{T_{cw} - T_f} \tag{A6}$$

$$\frac{\partial E}{\partial T_f} = \frac{M_w(T_{cw} - T_f) + M_w(T_f - T_w)}{(T_{cw} - T_f)^2} \tag{A7}$$

$$\frac{\partial E}{\partial T_{cw}} = -M_w \frac{T_f - T_w}{(T_{cw} - T_f)^2} \tag{A8}$$

It becomes evident that $\sigma_E$ is strongly related to the difference $T_f - T_{cw}$, which should be maximized. To achieve this, we aim to maximize the difference between $T_{cw}$ and $T_h$, while minimizing $M_{cw}$. However, in a hypothetical scenario with



$M_{cw} = 5$ g, $M_w = 495$ g, $T_{cw} = 273K$, $T_w = 373K$, and $E = 6.58$ g, the uncertainty $\sigma_E$ associated with $E$ is equal to 2.6 g, which is approximately $\frac{1}{3}$ of the value of $E$. Realistically, this experiment is challenging to conduct, primarily because it would be extremely difficult to maintain thermal equilibrium for 5 g of water with the internal wall. This was also recongized in Austin (1990).

     An alternative way for estimating the calorimetric constant $E$ is discussed in Fasani et al. (2023). Here the parameter E is re-
trieved using the same energy balance as in Eq. 1, and in detail inverting Eq. 3 expliciting $E$ as a function of $\theta_w^M$. This approach requires the employment of wet snow samples with known liquid water content, with all the relative associated difficulties in controlling the process. We present here the analysis on the uncertainty $\sigma_E$ associated to the parameter $E$ following the same approach as in Eq. 3.1. In detail, by inverting Eq. 3, we can obtain $E$ as:

$$E = \frac{(T_f - T_s)M_s + \frac{L}{C}(M_s - \theta_w^M V_s \rho_w)}{T_w - T_f} - M_w \tag{A9}$$

By analyzing the uncertainty propagation of the measurements on $E$ with the same approach presented in 3.1 we can retrieve the uncertainty on the estimation of E, $\sigma_E$:

$$\sigma_E = \sqrt{\left(\frac{\partial E}{\partial T_f}\right)^2 \sigma_{T_f}^2 + \left(\frac{\partial E}{\partial T_w}\right)^2 \sigma_{T_w}^2 + \left(\frac{\partial E}{\partial M_s}\right)^2 \sigma_{M_s}^2 + \left(\frac{\partial E}{\partial \theta_w^M}\right)^2 \sigma_{\theta_w^M}^2 + \left(\frac{\partial E}{\partial V_s}\right)^2 \sigma_{V_s}^2 + \left(\frac{\partial E}{\partial M_w}\right)^2 \sigma_{M_w}^2} \tag{A10}$$

The partial derivatives in Eq. (A10) are calculated as follows:

$$\frac{\partial E}{\partial T_f} = \frac{M_s(T_w - T_s) + \frac{L}{C}(M_s - \theta_w^M V_s \rho_w)}{(T_w - T_f)^2} \tag{A11}$$

$$\frac{\partial E}{\partial T_w} = -\frac{M_s(T_f - T_s) + \frac{L}{C}(M_s - \theta_w^M V_s \rho_w)}{(T_w - T_f)^2} \tag{A12}$$

$$\frac{\partial E}{\partial M_s} = \frac{T_f - T_s}{T_w - T_f} + \frac{L}{C(T_w - T_f)} \tag{A13}$$

$$\frac{\partial E}{\partial \theta_w^M} = -\frac{V_s \rho_w L}{C(T_w - T_f)} \tag{A14}$$

$$\frac{\partial E}{\partial V_s} = -\frac{\theta_w^M \rho_w L}{C(T_w - T_f)} \tag{A15}$$

$$\frac{\partial E}{\partial M_w} = -1 \tag{A16}$$





It is evident that $\sigma_E$ is strongly related to the difference $T_f - T_w$, which should be maximized. However, in a hypothetical scenario with $M_w = 167$ g, $M_s = 83$ g, $T_w = 313$K, the uncertainty $\sigma_E$ associated with $E$ is equal to 4.87 g, which is more than $\frac{2}{3}$ of the value of $E$, here set to 6.58 g.

A possible mitigation for the large error associated to the estimation of $E$, it is to use hot water and ice instead of water at different temperatures. However, it is again important to use the propagation of uncertainty to find how uncertainty is such kind of measurment of $E$. For the ice-water case, the heat-balance equation to retrieve $E$ becomes:

$$\underbrace{Q_{hot\ water}}_{M_w C(T_f - T_w)} + \underbrace{Q_{calorimeter}}_{M_{cal} C_{cal}(T_f - T_w)} + \underbrace{Q_{melting\ ice}}_{L M_i} + \underbrace{Q_{melted\ ice}}_{M_i C(T_f - T_{mi})} = 0 \tag{A17}$$

From where we can derive $E$ as follow:

$$E = \frac{M_i(C_i T_i - L - T_f)}{C(T_f - T_w)} - M_w \tag{A18}$$

Here, $M_i$ and $T_i$ represent the mass and temperature of ice, while $M_w$ and $T_w$ represent the mass and temperature of hot water. $T_{mi}$ instead is the temperature of the water coming from the melted ice cube, and it is equal to 273.15 K. Also for this case we can propagate the measurements uncertainty and find the associated $\sigma_E$

$$\sigma_E = \sqrt{\left(\frac{\partial E}{\partial M_i}\right)^2 \sigma_{M_i}^2 + \left(\frac{\partial E}{\partial M_h}\right)^2 \sigma_{M_h}^2 + \left(\frac{\partial E}{\partial T_h}\right)^2 \sigma_{T_h}^2 + \left(\frac{\partial E}{\partial T_f}\right)^2 \sigma_{T_f}^2 + \left(\frac{\partial E}{\partial T_i}\right)^2 \sigma_{T_i}^2} \tag{A19}$$

The partial derivatives in Eq. (A19) are calculated as follows:

$$\frac{\partial E}{\partial M_i} = \frac{C_i T_i - L - C T_f}{C(T_f - T_h)} \tag{A20}$$

$$\frac{\partial E}{\partial M_h} = -1 \tag{A21}$$

$$\frac{\partial E}{\partial T_h} = \frac{M_i(C_i T_i - L - C T_f)}{C(T_f - T_w)^2} \tag{A22}$$

$$\frac{\partial E}{\partial T_f} = \frac{-C M_i(T_f - T_h) - M_i(C_i T_i - L - C T_f)}{C(T_f - T_w)^2} \tag{A23}$$

$$\frac{\partial E}{\partial T_i} = \frac{M_i C_i}{C(T_f - T_w)} \tag{A24}$$





Analyzing the error for this case, we observe that it is strongly influenced by the difference between $T_f$ and $T_w$. In a
hypothetical scenario with $M_h = 300g$, $M_i = 200g$, $T_h = 373K$, $T_i = 244.5K$, and $E = 6.58g$, the uncertainty $\sigma_E$ associated
with $E$ is equal to 1.6 g, which is approximately $\frac{1}{4}$ of the value of $E$. This is still an high uncertainty, but unlike the water-water
case, this experiment does not suffer from the same limitations. Hence, if it is impossible to retrieve the exact mass and specific
heat of the calorimeter, the best way to estimate $E$ would be using hot water and ice. It is important to note that for both cases,
using ice and water at extreme temperatures would lead to noticeable heat dispersion in a short amount of time, making it
challenging to obtain exact temperature measurements.

Because of these reasons we destructively analyzed our calorimeters to estimate $E$ using Eq. (2). Fortunately this process
has to be done once, and other users can use $E$ provided in the paper i.e., $E = 6.58g$ given that they use the same calorimeter.

**Appendix B**

In this Appendix, we will show all the formulations partly presented in (Colbeck, 1978) that allows us to prove that the relative
uncertainty produced by the melting calorimeter on the estimation of the ice volume is one order of magnitude lower than
the relative uncertainty produced by the freezing calorimeter on the water volume. With this mathematical step we unlock the
melting calorimeter for being "inherently inaccurate".

We compute the relative uncertainty produced by the melting calorimeter on the estimation of the ice volume starting from
Eq. (3) in the main paper, written in terms of $V_i$:

$$V_i = \frac{C}{\rho_i L}[M_w(T_w - T_f) - M_s(T_f - T_s)] \tag{B1}$$

Note that we do not account for $E$ to be consistent with the Colbeck's formulation. Hence, we compute the partial derivatives
for each of the measured variables, by assuming $k = \frac{C}{L}$:

$$\begin{cases} \dfrac{\partial V_i}{\partial T_w} = \dfrac{k}{\rho_i} M_w \\ \dfrac{\partial V_i}{\partial T_f} = -\dfrac{k}{\rho_i}(M_w + M_s) \\ \dfrac{\partial V_i}{\partial M_w} = \dfrac{k}{\rho_i}(T_w - T_f) \\ \dfrac{\partial V_i}{\partial M_s} = -\dfrac{k}{\rho_i}(T_f - T_s) \end{cases} \tag{B2}$$

and calculate the relative uncertainty $\Sigma_{V_i}$ as sum of each term:

$$\Sigma_{V_i} = \left|\frac{\partial V_i}{\partial T_w}\right|\left|\frac{T_w}{V_i}\right|\left|\frac{dT}{T_w}\right| + \left|\frac{\partial V_i}{\partial T_f}\right|\left|\frac{T_f}{V_i}\right|\left|\frac{dT}{T_f}\right| + \left|\frac{\partial V_i}{\partial M_w}\right|\left|\frac{M_w}{V_i}\right|\left|\frac{dT}{M_w}\right| + \left|\frac{\partial V_i}{\partial M_s}\right|\left|\frac{M_s}{V_i}\right|\left|\frac{dT}{M_s}\right| \tag{B3}$$

By simplifying, we obtain:

$$\Sigma_{V_i} = \left|\frac{\partial V_i}{\partial T_w}\right|\left|\frac{dT}{V_i}\right| + \left|\frac{\partial V_i}{\partial T_f}\right|\left|\frac{dT}{V_i}\right| + \left|\frac{\partial V_i}{\partial M_w}\right|\left|\frac{dM}{V_i}\right| + \left|\frac{\partial V_i}{\partial M_s}\right|\left|\frac{dM}{V_i}\right| \tag{B4}$$



**Table B1.** Variable values used for the computation of the relative uncertainty produced by the melting calorimeter on the estimation of the ice volume.

| Variable | Value | Unit |
|---|---|---|
| C | $4.2 \cdot 10^3$ | $\mathrm{J\,kg^{-1}\,K^{-1}}$ |
| L | $3.34 \cdot 10^5$ | $\mathrm{J\,kg^{-1}}$ |
| $\rho_i$ | 917 | $\mathrm{kg\,m^{-3}}$ |
| $M_w$ | 1.067 | kg |
| $T_w$ | 313.15 | K |
| $T_f$ | 277.04 | K |
| $M_s$ | 0.5335 | kg |
| $T_s$ | 273.15 | K |

By considering the values reported in Table B1, we obtain $\Sigma_{V_i} = 0.007426\mathrm{m}^3$. The values in the table are obtained by considering a snow sample with $M_w = 2 \cdot M_s$, $V_s = 10^{-3}\mathrm{m}^3$, $V_i = 500 \cdot 10^{-6}\mathrm{m}^3$, $V_i = 75 \cdot 10^{-6}\mathrm{m}^3$ and $V_a = 425 \cdot 10^{-6}\mathrm{m}^3$.

Similarly, for the freeezing calorimeter we can compute the relative uncertainty on the estimation of the water volume starting from the equation of Jones, Eq. (6) in the main paper, written in terms of $V_W$:

$$V_w = \frac{M_o C_o(T_{fo} - T_o) + M_s C_s(T_{fo} - T_s)}{\rho_w L} \tag{B5}$$

Hence, we compute the partial derivatives for each of the measured variables, by assuming $k = \frac{C}{L}$:

$$\begin{cases} \dfrac{\partial V_w}{\partial T_o} = -\dfrac{M_o C_o}{\rho_w L} \\ \dfrac{\partial V_w}{\partial T_{fo}} = \dfrac{M_o C_o + M_s C_s}{\rho_w L} \\ \dfrac{\partial V_w}{\partial M_o} = \dfrac{C_o(T_{fo} - T_o)}{\rho_w L} \\ \dfrac{\partial V_w}{\partial M_s} = \dfrac{C_s(T_{fo} - T_s)}{\rho_w L} \end{cases} \tag{B6}$$

and calculate the relative uncertainty $\Sigma_{V_w}$ as sum of each term:

$$\Sigma_{V_w} = \left|\frac{\partial V_w}{\partial T_o}\right|\left|\frac{T_o}{V_w}\right|\left|\frac{dT}{T_o}\right| + \left|\frac{\partial V_w}{\partial T_{fo}}\right|\left|\frac{T_{fo}}{V_w}\right|\left|\frac{dT}{T_{fo}}\right| + \left|\frac{\partial V_w}{\partial M_o}\right|\left|\frac{M_o}{V_w}\right|\left|\frac{dM}{M_o}\right| + \left|\frac{\partial V_w}{\partial M_s}\right|\left|\frac{M_s}{V_w}\right|\left|\frac{dM}{M_s}\right| \tag{B7}$$

By simplifying, we obtain:

$$\Sigma_{V_w} = \left|\frac{\partial V_w}{\partial T_o}\right|\left|\frac{dT}{V_w}\right| + \left|\frac{\partial V_w}{\partial T_{fo}}\right|\left|\frac{dT}{V_w}\right| + \left|\frac{\partial V_w}{\partial M_o}\right|\left|\frac{dM}{V_w}\right| + \left|\frac{\partial V_w}{\partial M_s}\right|\left|\frac{dM}{V_w}\right| \tag{B8}$$

$dM$ and $dT$ are the instrumental uncertainties and are fixed to $10^{-4}$ g and 0.1 K respectively. By considering the values reported in Table B2, we obtain $\Sigma_{V_w} = 0.014809\mathrm{m}^3$ that results twice the error $\Sigma_{V_i}$ calculated for the melting calorimeter.



**Table B2.** Variable values used for the computation of the relative uncertainty produced by the freezing calorimeter on the estimation of the water volume.

| Variable | Value | Unit |
|----------|-------|------|
| $C_o$ | $1.83 \cdot 10^3$ | $\mathrm{J\,kg^{-1}\,K^{-1}}$ |
| $C_s$ | $2.09 \cdot 10^3$ | $\mathrm{J\,kg^{-1}\,K^{-1}}$ |
| L | $3.34 \cdot 10^5$ | $\mathrm{J\,kg^{-1}}$ |
| $\rho_w$ | 1000 | $\mathrm{kg\,m^{-3}}$ |
| $M_o$ | 0.69355 | kg |
| $T_o$ | 243.15 | K |
| $T_f o$ | 267.69 | K |
| $M_s$ | 0.5335 | kg |
| $T_s$ | 273.15 | K |

The values in the table are obtained by considering a snow sample with $M_o = 1.3 \cdot M_s$, $V_s = 10^{-3} \mathrm{m}^3$, $V_i = 500 \cdot 10^{-6} \mathrm{m}^3$, $V_i = 75 \cdot 10^{-6} \mathrm{m}^3$ and $V_a = 425 \cdot 10^{-6} \mathrm{m}^3$.

*Author contributions.* RB and CM designed the research, carried out the experiments, processed the data, and wrote the paper; VP helped with the mathematical formulation and wrote the Appendix with the relative codes; NC contributed with the field experiments; MB and FC
contributed to the WFJ data collection; all the authors contributed to the analysis and interpretation of the results and provided feedback on the final text.

*Competing interests.* The authors declare that they have no conflict of interest.

*Acknowledgements.* This work was supported by the joint project Swiss National Science Foundation (SNF) - Autonomous Province of Bolzano (Italy) "Snowtinel: Sentinel-1 SAR assisted catchment hydrology: toward an improved snow-melt dynamics for alpine regions"
Contract No. 200021L205190.



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
