# Peer review of "Unlocking the Potential of Melting Calorimetry: A Field Protocol for Liquid Water Content Measurement in Snow"

_EGUsphere, 2024_

## Referee Comment (RC2)

**General comment**:

This manuscript is a reappraisal of melting calorimetry for the measurement of liquid water in wet snow. Both melting and freezing calorimetry are compared. The work achieved is certainly valuable and worth of publication. However, the paper should be more concise. For instance, the part of Section 3 before Subsection 3.1 is not necessary because it is repeated later. And the figure captions should simply be descriptions of the figures needed for understanding.

**Main comments**:

My main concern is related to the key quantity, the volumetric liquid-water content, first mentioned in the abstract, and later at several places of the manuscript, e.g. Line 664. Its definition is the volume fraction of liquid water for a given test sample of snow, $\theta_v = V_w/V_s$, where $V_w$ is the volume of liquid water and $V_s$ is the volume of the snow. In Table 1, the respective quantity $\theta_\omega$ first appears as a percentage of liquid water "for snow volume", whatever this means. A few lines later in this table, $\theta_\omega$ appears as the mass fraction of liquid-water mass to total snow mass, independent of the snow volume. And this is the quantity required in the heat-budget equation (1).

To get the volumetric liquid-water content, $\theta_\omega$ must be multiplied with the ratio of snow density to density of liquid water. This ratio only reaches 1 when all snow is melted. Otherwise, it is smaller than 1. The dielectric sensors used today for the measurement of the liquid-water content are based on $\theta_v$, not $\theta_\omega$, see e.g. the intercomparison paper of Denoth et al. (1984.). It appears that the authors do not distinguish between the two quantities. And this is a mistake.

Please also note that "Mass of liquid-water fraction" (in Table 1, and near Equation (1)) is incorrect. A mass cannot be a fraction, because mass has units of kg, whereas a fraction is a number.

Another remark to Table 1 is to the description of the snow temperature, $T_s$. The given temperature is the melting temperature of pure ice, and indeed, this temperature is found throughout in wet snow (if salt or other ionic impurities are not involved). This value is not "by definition", but because water and ice are in good contact in wet snow, and heat conduction forces ice and water to be at the same temperature in wet snow.

**Small details**:

Line 107: clarifiy ... "technique accuracy"....

Line 118: correct ... "do not account not"...

Line 119: "Something that was never attempted in the past." Please be careful with such statements. You cannot be sure.

Line 143; ..."create an adiabatic environment, ensuring ideal heat exchange"... This sounds contradictive, because there is no heat exchange in adiabatic processes. Perhaps you mean that there is no heat exchange between the environment and the calorimeter.

Line 257: Change "The uncertainty... as the squared root" to "The uncertainty... as the square root"...

Line 430: Change "temperature spectrum" to "temperature range".

---

## Author Comment (AC1)

**Answer to the Referee #1 – Manuscript tc-2024-1708**

*I was unable to open the online resources for review. I tried twice and am not sure where the error is occurring so this is not currently part of my review.*

Currently, these resources are accessible upon request, which must be submitted via Google Colab. We apologize you were not granted access but for some reasons we did not receive any requests. However, to ensure wider accessibility, we plan to make these resources publicly available following the paper acceptance.

*The manuscript by Barella et al. describes a detailed analysis of errors in melt calorimetry for snow liquid water content (LWC) calculations. They compare this to that of freezing calorimetry and additionally conduct experiments to determine some of the random user uncertainties associated with the method. The authors present some melt calorimetry field protocol and conduct field investigations to show the utility of the methods.*

*Overall, I really like the concept of this study as I stated in my previous review of an earlier version of this manuscript. I think that this study adds a great contribution to the use and application of melt calorimetry. However, many of my primary concerns in the previous review have been addressed.*

We thank the review for his positive comment on the manuscript.

*I suggested that this may be more appropriate as a technical note somewhere since it is an advancement on existing methods and equations.*

Checking on the available manuscript type in TC there is no Technical Note (https://www.the-cryosphere.net/about/manuscript_types.html). Among all the manuscript types and according to the editorial rule we think that our work could fit to be published as a Research Article.

*Making the code available helps bring this towards a research article and the addition of the calorimeter constant and experiments are an excellent addition. This will greatly improve my own work and the manuscript rightfully points out the high errors resulting from methods published in my own work. However, in my opinion it needs to be re-organized to be a research article. I would like to emphasize that I really like the project and that these comments are meant to be constructive and helpful in producing a more readable and impactful final paper.*

We thank the reviewer for the positive comments. We will implement all the constructive comments in order to make the manuscript more readable and impactful.

*I think more background as to why LWC in snow is important could be included, as stated in my previous review. Here are some of those papers. This is a minor point that I wish the authors to re-consider. Examples of possible background papers for consideration:*

*Valence et al., 2022: https://tc.copernicus.org/articles/16/3843/2022/*

This paper is about the spatiotemporal monitoring of the snowpack due to ROS events.

*Donahue et al., 2022: https://tc.copernicus.org/articles/16/43/2022/*

This paper is about the LWC mapping at millimeter scale using optical measurements.

*Eiriksson et al., 2013: https://onlinelibrary.wiley.com/doi/10.1002/hyp.9666*

Paper about the importance of downslope flows during snowmelt and ROS events.
*Leroux et al., 2020: https://doi.org/10.1029/2020WR027466*

Paper about lateral flows modeling.

*Schlumpf et al., 2024: https://www.sciencedirect.com/science/article/pii/S0165232X23002872*

Mechanical properties of snow.

Even if all these papers are a great contribution to the field, they are beyond the scope of our paper. However, we understand the reviewer's suggestion to include additional papers demonstrating the broader implications of LWC knowledge, rather than classifying them as state-of-the-art or related work. While we already cite eight background studies, we will incorporate the suggested five papers as well.

*Major comments:*

- *The frequent mention of how significant of an improvement is made by this work makes it appear more like an argument/perspective manuscript. I think the tone of the writing could be improved. The math is solid and I think the work stands on it's own quite well without the need to argue for how much of an improvement on previous work. With how frequent this is done it lengthens some of the paragraphs and I think those familiar with the literature will see the impact of this improvement.*

We will revise the article to improve conciseness and readability. Unnecessary repetitions, which were added from the previous version to highlight the novelties, will be removed and the already consolidated concepts will be streamlined for clarity.

- *There is a lot going on and the writing jumps around quite a bit. I think that the manuscript needs to be re-organized for readability. In the current form, the writing is difficult to interpret what is methods, results, or discussion/interpretation. The manuscript would benefit from having distinct Methods, Results, and Discussion sections. As it is currently written, the manuscript bounces around quite a bit and is hard to follow.*

We appreciate the reviewer's feedback (this and the next minor ones) identifying the conflation of methods and results especially for Section 3.4. To address this, we will relocate section 3.4 (including dry snow and ice cube experiments) to the results portion of the paper. In this way, the revised structure will clearly delineate methods from results, enhancing overall readability. Specifically, Sections 3.1, 3.2, and 3.3, focusing on uncertainty derivation

and uncertainty analysis, will precede the protocol definition in Section 4. Subsequently, in Section 5 we will describe how the melting calorimeter is applied in the dry snow and ice cube experiment to random uncertainty identification (formerly Section 3.4), and LWC profile estimation for WFJ and Schnalstal (current Section 5, which will be significantly shortened according to your next comments). We believe that this rearrangement of the paper structure will improve the readability.

*More specific Comments (by line number):*

*326-328: This is one example of the type of statement that is repeated multiple times in the paper. Once the results are shown, the work stands on its own and this argument can be made once in a discussion or conclusion.*

We have deleted this repetition that was introduced to better grasp the novelty of the paper.

*457-465: This seems like the methods for the experiments, but not enough details are given to re-produce this work. Some details are given later, but still not quite enough.*

As stated before, we moved this section on the results section of the paper and this will address this concern.

*489: Were these conditions in the field or in a cold lab?*

It has been performed in a cold lab. We added this information.

*530: The current flow of the manuscript is odd. The protocol comes after discussion of the user experiments, even though much of the protocol is based on the uncertainty points made prior to the experiment sections. This is where a more clear methods, results, discussion could clarify much of this. The discussion of the uncertainty results would lead into the protocol discussion quite well.*

The new structure change described before will address this point.

*571: How did you dry the calorimeter? I assume using a rag or something, but perhaps you learned something that could be useful here to reduce uncertainty with any moisture left in the container.*

Any residual moisture will contribute to the measured mass and temperature of the hot water, parameters accounted for in the calculations. However, drying the external surface of the calorimeter and its lid is essential. Moisture on these surfaces is not factored into the heat exchange equation and can affect mass measurements. That said, our practical experience suggests that residual water droplets on the calorimeter exterior have negligible impact on mass measurements.

*581: The SSA and IR seem to add little to this paper that is so focused on calorimetry.*

We will use the same presenting scheme used for Schnalstal also for WFJ eliminating the IR and SSA information.

*591-598: This is a lot of details about the dates and max snow depth when only a single day was used. If only one of the 36 days were used, then only details from that one day are necessary.*

We shortened that section and focused on the detail of that single day's measurements. This helped to shorten the paper.

*638-641: I do not see the justification for not including the two depths that had different values. Yes, the LWC can vary significantly but then so could the values that you did compare. If the same volume of snow is not being tested and compared using the different methods, then all results should be used for comparison. Also, with 36 days you could have many more than 16 or 18 measurements for comparison that would be better for comparison statistically.*

We thank the reviewer for his comment, which provides an opportunity to clarify this point. As Boyne and Fisk (1987) reported, "the work was conducted in a laboratory cold room over a range of 0-14 mkg per 100 mkg of snow. The range is typical of a well-drained snowpack free of stratigraphic boundaries. Attempts to compare measurement methods in stratified snowpack have not been successful because of the spatial variability of liquid water". To ensure comparability with Boyne and Fisk (1987) results, we focused on similar experimental conditions, excluding the two measurements influenced by impermeable boundaries, which produced inconsistent results between the two methods in the WFJ profile. We agree that a more comprehensive statistical analysis would provide deeper insights into the comparison of the two methods. However, due to the paper's length constraints and the removal of detailed WFJ campaign data, as suggested by your previous comment, we believe such an analysis is beyond the current scope. Nevertheless, all data will be publicly accessible upon the relative publication, enabling future researchers to conduct in-depth statistical comparisons.

*672-673: "very similar" and "good stability" are not defined. At the previous site % values are given but not here. Please be consistent in quantitative comparisons.*

Thanks for pointing out this issue. We modified as follows:

"Finally, to verify the consistency of our measurements, we repeated a subset of measurements at very close distances on a uniform layer. The results of these repeated measurements yielded the following $\theta_w$ values: 7.27, 7.10, 7.19, and 7.20, with corresponding standard deviations of 0.50, 0.46, 0.56, and 0.52 respectively. These values were found to be very similar and fell within the uncertainty range, demonstrating the good stability of the melting calorimetry technique. Additionally, a z-test for null hypothesis verification (Rouder et al., 2009), applied at the difference between these values confirmed the evidence."

*967: Please use the appropriate reference as given in my previous review: Webb et al., 2021: https://www.mdpi.com/2072-4292/13/22/4617*

We updated the reference

---

## Author Comment (AC2)

**Answer to the Referee #2 – Manuscript tc-2024-1708**

*General comment:*
*This manuscript is a reappraisal of melting calorimetry for the measurement of liquid water in wet snow. Both melting and freezing calorimetry are compared. The work achieved is certainly valuable and worth of publication.*

We thank the reviewer for his thoughtful comments and valuable suggestions, which will undoubtedly enhance the quality of our manuscript. Our responses are reported in blue in the following.

*However, the paper should be more concise. For instance, the part of Section 3 before Subsection 3.1 is not necessary because it is repeated later. And the figure captions should simply be descriptions of the figures needed for understanding.*

We agree that the paper should be more concise. Interestingly, our original intent was a short communication that addressed two key points clarifying the Colbeck's claims about the calorimeter inaccuracy and providing a clear field protocol for consistent measurements. However, as we delved deeper using the calorimeter, we uncovered broader issues related to its use in existing literature. This led us to create a more in-depth review and potential correction of the current state of the art for LWC estimation using calorimetry. With this round of revision we aimed at eliminating redundancies, avoiding repetition, simplifying figure captions, and reorganizing some sections (see responses to review 1) of the paper for improved readability. This allowed us to reduce the length of the manuscript and make it more concise.

*Main comments:*
*My main concern is related to the key quantity, the volumetric liquid-water content, first mentioned in the abstract, and later at several places of the manuscript, e.g. Line 664. Its definition is the volume fraction of liquid water for a given test sample of snow, $\theta_v = V_v/V_s$, where $V_v$ is the volume of liquid water and $V_s$ is the volume of the snow. In Table 1, the respective quantity $\theta_w$ first appears as a percentage of liquid water "for snow volume", whatever this means. A few lines later in this table, $\theta_w$ appears as the mass fraction of liquid-water mass to total snow mass, independent of the snow volume. And this is the quantity required in the heat-budget equation (1). To get the volumetric liquid-water content, $\theta_w$ must be multiplied with the ratio of snow density to density of liquid water. This ratio only reaches 1 when all snow is melted. Otherwise, it is smaller than 1. The dielectric sensors used today for the measurement of the liquid-water content are based on $\theta_v$, , not $\theta_w$, see e.g. the intercomparison paper of Denoth et al. (1984.). It appears that the authors do not distinguish between the two quantities. And this is a mistake.*

We thank the reviewer for identifying this error that could have caused significant confusion. This oversight escaped our initial proofreading. The correct form in table I and L185 is

$M_{W\theta\omega} = \theta_\omega M_s \rho_s / \rho_w$ . In general, we are following the for all the our notation the one proposed by the international classification of snow, where $\theta_w$ is used for both volume and mass (see Appendix D pag. 53). Indeed, we recognize there are two main definitions adopted: one using the mass (e.g., CROCUS, calorimetry) and one using the volume (e.g., SNOWPACK, dielectric probes). Interestingly enough some years ago we started a discussion on how SMRT should specify the liquid water content within the community, since it was specified with the fractional volume of water with respect to ice, a convenient quantity in electromagnetism, but not for a snow scientist. These are all "liquid water content" but expressed in different units. While converting between the different definitions is straightforward, distinct symbols should be used for clarity. Additionally, it is crucial to consider how errors propagate under each definition. Since our research primarily focuses on the volume fraction of liquid water within snow, particularly due to our interest in radar and dielectric sensors, we consequently expressed $\theta_w$ in volume fraction (%) in this paper. We have adopted the notation recommended by the international classification of snow for consistency i.e., $\theta_w$ but it has to be distinguished from the percentage of mass liquid water content. Thank you for highlighting this crucial point, as it could have led to misunderstandings.

*Please also note that "Mass of liquid-water fraction" (in Table 1, and near Equation (1)) is incorrect. A mass cannot be a fraction, because mass has units of kg, whereas a fraction is a number.*

Thanks for pointing out this. We indeed meant the "percentage of mass liquid water content" but somehow we contract this long form in an incorrect way. This will be changed.

*Another remark to Table 1 is to the description of the snow temperature, Ts. The given temperature is the melting temperature of pure ice, and indeed, this temperature is found throughout in wet snow (if salt or other ionic impurities are not involved). This value is not "by definition", but because water and ice are in good contact in wet snow, and heat conduction forces ice and water to be at the same temperature in wet snow.*

We appreciate the reviewer for highlighting this critical point. By stating "by definition," we intended to emphasize that our calorimetric formula is strictly applicable to wet snow at 0°C. Our aim with the "by definition" statement was to caution users about the calorimetric applicability to prevent invalid measurements (e.g., Mavrovic et al. (2020)). We will clarify this point further in the revised manuscript and delete "by definition".

*Small details:*
*Line 107: clarifiy ... "technique accuracy"....*

Thank you for pointing out this issue, we will change the sentence as: "This provides a more rigorous understanding of the technique reliability, quantifying the measurement uncertainty, so that the application of melting calorimetry in the future is correct and sound from the critics".

*Line 118: correct ... "do not account not"...*

Thanks for pointing out this error. Corrected.

*Line 119: "Something that was never attempted in the past." Please be careful with such statements. You cannot be sure.*

Thanks for pointing out this issue. This is to our knowledge. We removed this statement as suggested.

*Line 143; ..."create an adiabatic environment, ensuring ideal heat exchange"... This sounds contradictive, because there is no heat exchange in adiabatic processes. Perhaps you mean that there is no heat exchange between the environment and the calorimeter.*

Thanks for pointing out this aspect that may generate confusion. We change this sentence as follows: "The calorimeter is designed as an insulated container to maintain a given temperature and create an insulated environment from the outside, ensuring ideal heat exchange between the snow sample and the melting or freezing agent".

*Line 257: Change "The uncertainty... as the squared root" to "The uncertainty... as the square root"...*

Thanks. Modified.

*Line 430: Change "temperature spectrum" to "temperature range".*

Changed.

---

## Author Comment (AC3)

**Errata corrige - Answer to the Referee #2 – Manuscript tc-2024-1708**

We just realized that we wrongly copied one of the formulae inside our response. To avoid other possible error and misunderstanding, here a screenshot of how Table I reads in our new version of the manuscript

| | |
|---|---|
| $M_{W_{\theta_w}}$ | Mass of the liquid water inside the sample, it can be expressed as: $M_{W_{\theta_w}} = \theta_w M_s \frac{\rho_w}{\rho_s}$ |

We apologize for any inconvenience this may have cause.

---

## Author Response (AR2)

**Review answers Manuscript tc-2024-1708**

**Answer to the Referee #1**

*Lines 518-520: The previous version stated what layers were not included in the analysis. The revised version is misleading. In a natural snowpack, more recent studies than Boyne and Fisk show that ice layers are not often impermeable as lab studies have suggested. Furthermore, other layer interfaces may be more effective at retaining water resulting in the high variability of LWC (Techel and Pielmeier, 2011 is a great example). While the authors try to justify the exclusion of comparisons in the field with the Boyne and Fisk laboratory study. I disagree. All of the data for comparison must be included in the mean difference and standard deviation %s shown.*

Thank you for your comment, we appreciate the opportunity to further clarify this important point. As noted by Boyne and Fisk (1987), *"Attempts to compare measurement methods in stratified snowpack have not been successful because of the spatial variability of liquid water [Denoth et al., 1984; Boyne and Georte, 1987].".* To mitigate this, and following the general procedure used by Boyne and Fisk (1987), we excluded sampling points where conditions of LWC were likely inconsistent (as is happening for the two excluded points). While our paper already discussed the differences between Dentohmeter and calorimetric measurements (comparing also with previous studies e.g., Perla 1991), the primary goal of this exercise was to indirectly compare our findings with those of Boyne and Fisk (1987), so we need to be sure to undergo to the same hypothesis. To enhance clarity, we have revised the sentence as follows:

"Following the procedure outlined in Boyne and Fisk (1987), we analyzed 16 measurements where both instruments sampled the same LWC conditions for the profile shown in Fig. 9, excluding the measurements at H=87 and H=0 cm. These points were omitted due to the high horizontal variability in LWC, which made it impossible to ensure that we were sampling identical conditions (see *Techel and Pielmeier, 2011*). The results showed a mean difference of 0.96% and a standard deviation of 1% between the two methods, aligning closely with previous findings for alcohol calorimeters and Denothmeters."

*Lines 546-547: Please remove this statement. I think the preference of one method over another is not justified with only two pits of similar conditions. Much more data and a third method for comparison would be necessary to make this claim.*

Even though we think that our previous statement was already expressed in a neutral form we further soften our statement as: "Although the coherence of the profile along the vertical axis and its correlation with density and stratigraphy are in favor of calorimetric measurements, additional data and analysis are required to fully support the preference of the calorimetry profile over the Denothmeter."

*Other minor concerns to be addressed:*

*Line 414: How did you dry the calorimeter and what insights do you have? The authors gave a great response in the reviewer response document, but nothing was added to the manuscript which was the intent of the question.*

Considering our previous answer, we modified the text in this way: "13. Empty and dry the calorimeter thoroughly before subsequent measurements. While carefully drying the calorimeter is advised, the residual water inside the calorimeter will be included in the mass and temperature measurement of the hot water, whereas the potential water or snow attached on the outside part of the calorimeter has to be clean out (even though their mass is in general negligible)."

*Lines 607-609: This is a misleading statement as Webb et al. (2021) did compensate for energy losses to the calorimeter, albeit not using a calorimeter constant, the compensation was still applied. The present manuscript is certainly a better way of doing it than Webb et al. (2021), but they did address it.*

Thank you for the feedback. However, our statement was specifically addressing the use of a calorimetric constant to compensate for the factors influencing the melting process inside the calorimeter, rather than the compensation for thermal dispersions from the calorimeter with the external environment. While Webb et al. (2021) discussed in the section "Uncertainty and Future Work" about the potential energy losses, they do not appear to incorporate a compensation for the energy exchange with the internal wall of the calorimeter in their formulation.